# LINEAR MODE CONNECTIVITY IN DIFFERENTIABLE TREE ENSEMBLES

**Ryuichi Kanoh**[1,2], **Mahito Sugiyama**[1,2]
[1]National Institute of Informatics
[2]The Graduate University for Advanced Studies, SOKENDAI
{kanoh, mahito}@nii.ac.jp

## ABSTRACT

*Linear Mode Connectivity* (LMC) refers to the phenomenon that performance remains consistent for linearly interpolated models in the parameter space. For independently optimized model pairs from different random initializations, achieving LMC is considered crucial for understanding the stable success of the non-convex optimization in modern machine learning models and for facilitating practical parameter-based operations such as model merging. While LMC has been achieved for neural networks by considering the permutation invariance of neurons in each hidden layer, its attainment for other models remains an open question. In this paper, we first achieve LMC for *soft tree ensembles*, which are tree-based differentiable models extensively used in practice. We show the necessity of incorporating two invariances: *subtree flip invariance* and *splitting order invariance*, which do not exist in neural networks but are inherent to tree architectures, in addition to permutation invariance of trees. Moreover, we demonstrate that it is even possible to exclude such additional invariances while keeping LMC by designing *decision list*-based tree architectures, where such invariances do not exist by definition. Our findings indicate the significance of accounting for architecture-specific invariances in achieving LMC.

## 1 INTRODUCTION

A non-trivial empirical characteristic of modern machine learning models trained using gradient methods is that models trained from different random initializations could achieve nearly identical performance, even though their parameter representations differ. This empirical phenomenon can be understood if the outcomes of all training sessions converge to the same local minima. However, considering the complex non-convex nature of the loss surface, the optimization results are unlikely to converge to the same local minima. In recent years, particularly within the context of neural networks, the transformation of model parameters while preserving functional equivalence has been explored by considering the *permutation invariance* of neurons in each hidden layer (Hecht-Nielsen, 1990; Chen et al., 1993). Notably, only a slight performance degradation has been observed when using weights derived through linear interpolation between permuted parameters obtained from different training processes (Entezari et al., 2022; Ainsworth et al., 2023). This demonstrates that the trained models reside in different, yet equivalent, local minima. This situation is referred to as *Linear Mode Connectivity* (LMC) (Frankle et al., 2020). From a theoretical perspective, LMC is crucial for understanding the stable and successful application of non-convex optimization. As noted by Entezari et al. (2022) and Ainsworth et al. (2023), achievement of LMC suggests that loss landscapes often contain (nearly) a single basin after accounting for all possible invariances, which can be an intuitive reason for the robustness of gradient methods to different random initialization and data batch orders. In addition, LMC also holds significant practical importance, enabling techniques such as model merging (Wortsman et al., 2022; Ortiz-Jimenez et al., 2023) by weight-space parameter averaging.

Although neural networks have been studied most extensively studied among the models trained using gradient methods, other models also thrive in real-world applications. A representative is decision tree ensemble models, such as random forests (Breiman, 2001). A decision tree ensemble makes predictions by combining the outputs of multiple trees that recursively split the data into subsets at each node and make final predictions at their leaves. While they are originally trained by greedy

algorithms, not by gradient algorithms, their *differentiable* variant — called *soft tree ensembles*, which learn parameters of the entire model through gradient-based optimization — have recently been actively studied. Not only empirical studies regarding accuracy and interpretability (Popov et al., 2020; Hazimeh et al., 2020; Chang et al., 2022), but also theoretical analyses have been performed (Kanoh & Sugiyama, 2022; 2023). Moreover, the differentiability of soft trees allows for integration with various deep learning methodologies, including fine-tuning (Ke et al., 2019), dropout (Srivastava et al., 2014), and various stochastic gradient descent methods (Kingma & Ba, 2015; Foret et al., 2021). Furthermore, the soft tree represents the most elementary form of a hierarchical mixture of experts (Jordan & Jacobs, 1993). Investigating soft tree models not only advances our understanding of this particular structure but also contributes to broader research on the key components that are essential to developing large-scale models (Jiang et al., 2024).

A research question that we tackle in this paper is: "Can LMC be achieved for soft tree ensembles?". While achieving LMC has advanced the understanding of non-convex optimization and the use of model merging in neural networks, it has yet to be explored in tree ensemble models. The reasons behind achieving LMC, even in neural networks, are not fully understood, and it is also unclear whether LMC can be realized in soft tree ensembles, given their distinct architectures. Thus, our contribution of examining LMC in soft tree ensembles provides not only novel insights and techniques for tree ensemble models but also broadens the understanding of the LMC phenomenon by introducing perspectives beyond neural networks for the first time.

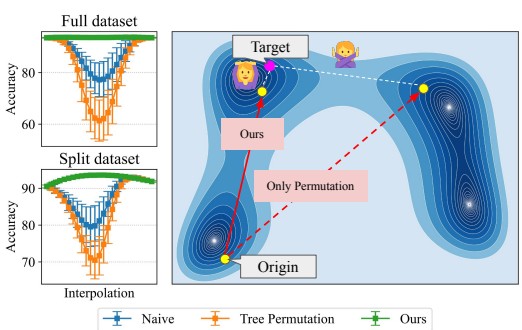

Figure 1: A representative experimental result on the MiniBooNE (Roe, 2010) dataset (left) and conceptual diagram of LMC for tree ensembles (right).

Our results, which are highlighted with a green line in the top left panel of Figure 1, clearly show that the answer to our research question is "Yes". This plot shows the variation in test accuracy when interpolating weights of soft oblivious trees, perfect binary soft trees with shared parameters at each depth, trained from different random initializations. The green line is obtained by the method introduced in this paper, where there is almost zero performance degradation. Furthermore, as shown in the bottom left panel of Figure 1, the performance can even improve when interpolating between models trained on split datasets.

The key insight is that, when performing interpolation between two model parameters, considering only tree permutation invariance, which corresponds to the permutation invariance of neural networks, is *not sufficient* to achieve LMC, as shown in the orange lines in the plots. An intuitive understanding of this situation is also illustrated in the right panel of Figure 1. To achieve LMC, that is, the green lines, we show that two additional invariances beyond tree permutation, *subtree flip invariance* and *splitting order invariance*, which inherently exist for tree architectures, should be accounted for.

Moreover, we demonstrate that it is possible to exclude such additional invariances while preserving LMC by modifying tree architectures. We realize such an architecture based on *a decision list*, a binary tree structure where branches extend in only one direction. By designating one of the terminal leaves as an empty node, we introduce a customized decision list that omits both subtree flip invariance and splitting order invariance, and empirically show that this can achieve LMC by considering only tree permutation invariance. Since incorporating additional invariances is computationally expensive, we can efficiently perform model merging on our customized decision lists.

Our contributions are summarized as follows:

- First achievement of LMC for tree ensembles with additional invariances beyond tree permutation.
- Development of a decision list-based architecture that does not involve additional invariances.
- A thorough empirical investigation of LMC across various tree architectures and real-world datasets.

## 2 PRELIMINARY

We prepare the basic concepts of LMC and soft tree ensembles.

## 2.1 LINEAR MODE CONNECTIVITY

Let us consider two models, $A$ and $B$, which have the same architecture. In the context of evaluating LMC, the concept of a "barrier" is frequently used (Ainsworth et al., 2023; Guerrero Peña et al., 2023). Let $\boldsymbol{\Theta}_A$ and $\boldsymbol{\Theta}_B$ be the parameters of models $A$ and $B$, respectively. Their shapes can be defined arbitrarily. In this paper, for tree ensemble models, $\boldsymbol{\Theta}_A, \boldsymbol{\Theta}_B \in \mathbb{R}^{M \times P}$ for the number $P$ of parameters per tree and the number $M$ of trees. Assume that $\mathcal{C} : \mathbb{R}^{M \times P} \to \mathbb{R}$ measures the performance of the model, such as accuracy. If higher values of $\mathcal{C}(\cdot)$ mean better performance, the barrier between two parameter vectors $\boldsymbol{\Theta}_A$ and $\boldsymbol{\Theta}_B$ is defined as:

$$\mathcal{B}(\boldsymbol{\Theta}_A, \boldsymbol{\Theta}_B) = \sup_{\lambda \in [0,1]} \left[ \lambda \mathcal{C}(\boldsymbol{\Theta}_A) + (1 - \lambda)\mathcal{C}(\boldsymbol{\Theta}_B) - \mathcal{C}(\lambda \boldsymbol{\Theta}_A + (1 - \lambda)\boldsymbol{\Theta}_B) \right]. \quad (1)$$

We can simply reverse the subtraction order if lower values of $\mathcal{C}(\cdot)$ mean better performance like loss.

Several techniques have been developed to reduce barriers by transforming parameters while preserving functional equivalence. Two main approaches are *activation matching* (AM) and *weight matching* (WM). AM takes the behavior of model inference into account, while WM simply compares two models using their parameters. The validity of both AM and WM has been theoretically supported by Zhou et al. (2023). Numerous algorithms are available for implementing AM and WM. For instance, Ainsworth et al. (2023) used a formulation based on the Linear Assignment Problem (LAP), also known as finding the minimum-cost matching in bipartite graphs, to determine suitable permutations. Guerrero Peña et al. (2023) employed a differentiable formulation that allows for the optimization of permutations using gradient-based methods.

Existing research has focused exclusively on neural networks such as multi-layer perceptrons (MLP) and convolutional neural networks (CNN). No studies have been conducted for soft tree ensembles.

## 2.2 SOFT TREE ENSEMBLE

Unlike typical hard decision trees, which explicitly determine the data flow to the right or left at each splitting node, soft trees represent the proportion of data flowing to the right or left as continuous values between 0 and 1. This approach enables a differentiable formulation. We use a sigmoid function, $\sigma : \mathbb{R} \to (0, 1)$ to formulate a function $\mu_{m,\ell}(\boldsymbol{x}_i, \boldsymbol{w}_m, \boldsymbol{b}_m) : \mathbb{R}^F \times \mathbb{R}^{F \times \mathcal{N}} \times \mathbb{R}^{1 \times \mathcal{N}} \to (0, 1)$ that represents the proportion of the $i$th data point $\boldsymbol{x}_i$ flowing to the $\ell$th leaf of the $m$th tree as a result of soft splittings:

$$\mu_{m,\ell}(\boldsymbol{x}_i, \boldsymbol{w}_m, \boldsymbol{b}_m) = \prod_{n=1}^{\mathcal{N}} \underbrace{\sigma(\boldsymbol{w}_{m,n}^\top \boldsymbol{x}_i + b_{m,n})}_{\text{flow to the left}}^{\mathbb{1}_{\ell \swarrow n}} \underbrace{(1 - \sigma(\boldsymbol{w}_{m,n}^\top \boldsymbol{x}_i + b_{m,n}))}_{\text{flow to the right}}^{\mathbb{1}_{n \searrow \ell}}, \quad (2)$$

where $\mathcal{N}$ denotes the number of splitting nodes in each tree. The parameters $\boldsymbol{w}_{m,n} \in \mathbb{R}^F$ and $b_{m,n} \in \mathbb{R}$ correspond to the feature selection mask and splitting threshold value for $n$th node in a $m$th tree, respectively. The expression $\mathbb{1}_{\ell \swarrow n}$ (resp. $\mathbb{1}_{n \searrow \ell}$) is an indicator function that returns 1 if the $\ell$th leaf is positioned to the left (resp. right) of a node $n$, and 0 otherwise.

If parameters are shared across all splitting nodes at the same depth, such perfect binary trees are called *oblivious trees*. Mathematically, $\boldsymbol{w}_{m,n} = \boldsymbol{w}_{m,n'}$ and $b_{m,n} = b_{m,n'}$ for any nodes $n$ and $n'$ at the same depth in an oblivious tree. Oblivious trees can significantly reduce the number of parameters from an exponential to a linear order of the tree depth, and they are actively used in practice (Popov et al., 2020; Chang et al., 2022).

To classify $C$ categories, the output of the $m$th tree is computed by the function $f_m : \mathbb{R}^F \times \mathbb{R}^{F \times \mathcal{N}} \times \mathbb{R}^{1 \times \mathcal{N}} \times \mathbb{R}^{C \times \mathcal{L}} \to \mathbb{R}^C$ as sum of the leaves $\boldsymbol{\pi}_{m,\ell}$ weighted by the outputs of $\mu_{m,\ell}(\boldsymbol{x}_i, \boldsymbol{w}_m, \boldsymbol{b}_m)$:

$$f_m(\boldsymbol{x}_i, \boldsymbol{w}_m, \boldsymbol{b}_m, \boldsymbol{\pi}_m) = \sum_{\ell=1}^{\mathcal{L}} \mu_{m,\ell}(\boldsymbol{x}_i, \boldsymbol{w}_m, \boldsymbol{b}_m)\boldsymbol{\pi}_{m,\ell}, \quad (3)$$

where $\mathcal{L}$ is the number of leaves in a tree. To facilitate understanding, the formulation for tree depth is $D = 1$ is illustrated in Figure 2.

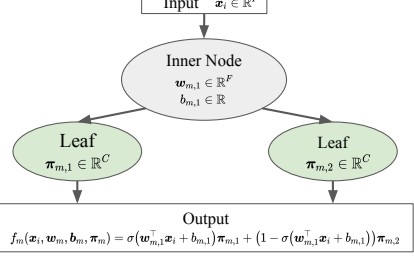

Figure 2: A soft decision tree with a single inner node and two leaf nodes.

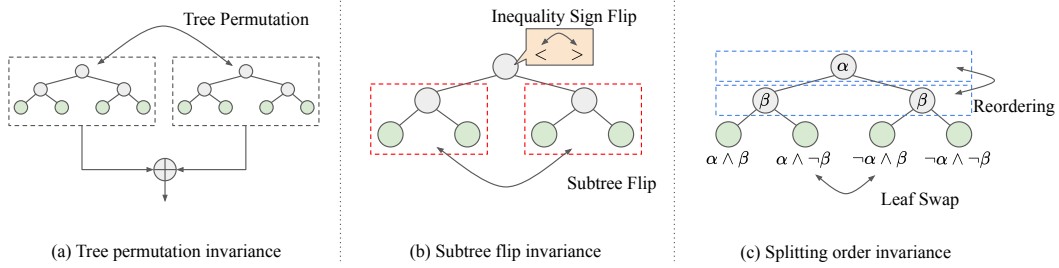

Figure 3: Invariances inherent to tree ensembles.

If $\mu_{m,\ell}(\boldsymbol{x}_i, \boldsymbol{w}_m, \boldsymbol{b}_m)$ takes the value of $1.0$ for one leaf and $0.0$ for the others, the leaf value itself becomes the prediction output, making the model behavior equivalent to that of a standard oblique decision tree (Murthy et al., 1994).

By combining this function for $M$ trees, we realize the function $f : \mathbb{R}^F \times \mathbb{R}^{M \times F \times \mathcal{N}} \times \mathbb{R}^{M \times 1 \times \mathcal{N}} \times \mathbb{R}^{M \times C \times \mathcal{L}} \to \mathbb{R}^C$ as an ensemble model consisting of $M$ trees:

$$f(\boldsymbol{x}_i, \boldsymbol{w}, \boldsymbol{b}, \boldsymbol{\pi}) = \sum_{m=1}^{M} f_m(\boldsymbol{x}_i, \boldsymbol{w}_m, \boldsymbol{b}_m, \boldsymbol{\pi}_m), \tag{4}$$

with the trainable parameters $\boldsymbol{w} = (\boldsymbol{w}_1, \ldots, \boldsymbol{w}_M)$, $\boldsymbol{b} = (\boldsymbol{b}_1, \ldots, \boldsymbol{b}_M)$, and $\boldsymbol{\pi} = (\boldsymbol{\pi}_1, \ldots, \boldsymbol{\pi}_M)$ being randomly initialized.

As shown in Equation 4, tree ensembles exhibit permutation invariance when the order of the $M$ trees is rearranged, which is similar to the permutation invariance observed in the hidden neurons of neural networks. However, as discussed in the next section, tree ensembles exhibit several other types of invariance beyond permutation, setting their behavior apart from that of neural networks. In addition to these invariances, there are several key differences between tree ensembles and neural networks. Due to the hierarchical binary tree structure, the influence of each node parameter on the overall model depends on its node position. Moreover, unlike neural networks, tree ensembles lack the concept of activation and intermediate layers. These factors make it challenging to directly apply the matching strategies used for neural networks to achieve LMC.

## 3 INVARIANCES INHERENT TO TREE ENSEMBLES

In this section, we discuss additional invariances inherent to trees (Section 3.1) and introduce a matching strategy specifically for tree ensembles (Section 3.2). We also show that the presence of additional invariances varies depending on the tree structure, and we present tree structures where no additional invariances beyond tree permutation exist (Section 3.3).

### 3.1 PARAMETER MODIFICATION PROCESSES

When we consider perfect binary trees, there are three types of invariance:

- **Tree permutation invariance.** In Equation 4, the behavior of the function does not change even if the order of the $M$ trees is altered, as shown in Figure 3(a). This corresponds to the permutation of hidden neurons in neural networks, which has been a subject of previous studies on LMC.

- **Subtree flip invariance.** When the left and right subtrees are swapped simultaneously with the inversion of the inequality sign at the split, the functional behavior remains unchanged, which we refer to *subtree flip invariance*. Figure 3(b) presents a schematic diagram of this invariance, which is not found in neural networks but is unique to binary tree-based models. Since $\sigma(-c) = 1 - \sigma(c)$ for $c \in \mathbb{R}$ due to the symmetry of $\mathrm{sigmoid}$, the inversion of the inequality is achieved by inverting the signs of $\boldsymbol{w}_{m,n}$ and $b_{m,n}$. Yadav et al. (2023) also focused on the sign of weights, but in a different way from ours. They paid attention to the amount of change from the parameters at the start of fine-tuning, rather than discussing the sign of the parameters.

- **Splitting order invariance.** Oblivious trees share parameters at the same depth, which means that the decision boundaries are straight lines without any bends. With this characteristic, even if the splitting rules at different depths are swapped, functional equivalence can be achieved if the positions of leaves are also swapped appropriately as shown in Figure 3(c). This invariance does not exist for non-oblivious perfect binary trees without parameter sharing, as the behavior of the decision boundary varies depending on the splitting order.

Note that MLPs also have an additional invariance beyond just permutation. Particularly in MLPs that employ ReLU as an activation function, the output of each layer changes linearly with a zero crossover. Therefore, it is possible to modify parameters without changing functional behavior by multiplying the weights in one layer by a constant and dividing the weights in the previous layer by the same constant. However, since the soft tree is based on the sigmoid function, this invariance does not apply. Previous studies (Entezari et al., 2022; Ainsworth et al., 2023; Guerrero Peña et al., 2023) have consistently achieved significant reductions in barriers without accounting for this scale invariance. This could be because changes in parameter scale are unlikely due to the nature of optimization via gradient descent. Conversely, when we consider additional invariances inherent to trees, the scale is equivalent to the original parameters.

## 3.2 Matching Strategy

When considering subtree flip invariance and splitting order invariance, it is necessary to compare multiple functionally equivalent trees and select the most suitable one for achieving LMC. Although comparing tree parameters is a straightforward approach, since the contribution of all the parameters in a tree is not equal, we apply appropriate weighting for each node. By interpreting a tree as a rule set with shared parameters as shown in Figure 4, we determine the weight of each splitting node by counting the

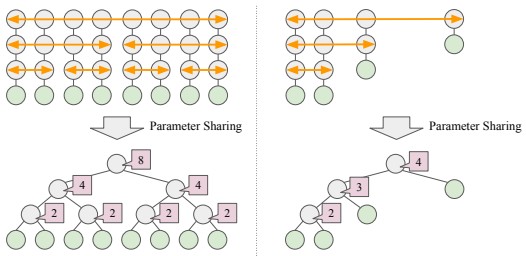

Figure 4: Weighting strategy.

number of leaves to which the node affects. For example, in the case of the example in the left-hand side of Figure 4, the root node affects eight leaves, nodes at depth 2 affect four leaves, and nodes at depth 3 affect two leaves. This strategy can apply to even trees other than perfect binary trees. For example, in the right example of Figure 4, the root node affects four leaves, a node at depth 2 affects three leaves, and a node at depth 3 affects two leaves.

Using the weighting operation described above, we present the straightforward matching procedure in Algorithms 1 and 2. We performed an exhaustive search to explore all patterns with subtree flip invariance and splitting order invariance, while handling tree permutation invariance with the LAP. We treat the output of each individual tree like the activation value of a neural network in the case of AM. Note that although it is necessary to solve the LAP multiple times for each layer in MLPs to perform coordinate descent (Ainsworth et al., 2023), tree ensembles require only a single run of the LAP since there is no concept of intermediate layers.

**Notations used in Algorithms 1 and 2.** Multidimensional array elements are accessed using square brackets [·]. For example, for $\boldsymbol{G} \in \mathbb{R}^{I \times J}$, $\boldsymbol{G}[i]$ refers to the $i$th slice along the first dimension, and $\boldsymbol{G}[:, j]$ refers to the $j$th slice along the second dimension, with sizes $\mathbb{R}^J$ and $\mathbb{R}^I$, respectively. Furthermore, it can also accept a vector $\boldsymbol{v} \in \mathbb{N}^l$ as an input. In this case, $\boldsymbol{G}[\boldsymbol{v}] \in \mathbb{R}^{l \times J}$. The FLATTEN function converts multidimensional input into a one-dimensional vector format. As the LINEARSUMASSIGNMENT function, scipy.optimize.linear_sum_assignment[1] is used to solve the LAP. In the ADJUSTTREE function, the parameters of a tree are modified according to the $u$th pattern among the enumerated $U \in \mathbb{N}$ total additional invariances patterns. Additionally, in the WEIGHTING function, parameters are multiplied by the square root of their weights to simulate the process of assessing a rule set. If the first argument for the UPDATEBESTOPERATION function, the input inner product, is larger than any previously input inner product values, then $u'$ is updated with $u$, the second argument. If not, $u'$ remains unchanged.

---

[1] https://docs.scipy.org/doc/scipy/reference/generated/scipy.optimize.linear_sum_assignment.html

---

**Algorithm 1:** Activation matching for soft tree ensembles

---

1   ACTIVATIONMATCHING($\boldsymbol{\Theta}_A \in \mathbb{R}^{M \times P}$, $\boldsymbol{\Theta}_B \in \mathbb{R}^{M \times P}$, $\boldsymbol{x}_{\text{sampled}} \in \mathbb{R}^{F \times N_{\text{sampled}}}$)

2      Initialize $\boldsymbol{O}_A \in \mathbb{R}^{M \times N_{\text{sampled}} \times C}$ and $\boldsymbol{O}_B \in \mathbb{R}^{M \times N_{\text{sampled}} \times C}$ to store outputs

3      **for** $m = 1$ *to* $M$ **do**

4          **for** $i = 1$ *to* $N_{\text{sampled}}$ **do**

5              Set the output of the $m$th tree with $\boldsymbol{\Theta}_A[m]$ using $\boldsymbol{x}_{\text{sampled}}[:, i]$ to $\boldsymbol{O}_A[m, i]$.

6              Set the output of the $m$th tree with $\boldsymbol{\Theta}_B[m]$ using $\boldsymbol{x}_{\text{sampled}}[:, i]$ to $\boldsymbol{O}_B[m, i]$.

7      Initialize similarity matrix $\boldsymbol{S} \in \mathbb{R}^{M \times M}$

8      **for** $m_A = 1$ *to* $M$ **do**

9          **for** $m_B = 1$ *to* $M$ **do**

10              $\boldsymbol{S}[m_A, m_B] \leftarrow$ FLATTEN($\boldsymbol{O}_A[m_A]$) $\cdot$ FLATTEN($\boldsymbol{O}_B[m_B]$)

11      $\boldsymbol{p} \leftarrow$ LINEARSUMASSIGNMENT($\boldsymbol{S}$)            // $\boldsymbol{p} \in \mathbb{N}^M$: Optimal assignments

12      $\boldsymbol{\Theta}_A, \boldsymbol{\Theta}_B \leftarrow$ WEIGHTING($\boldsymbol{\Theta}_A, \boldsymbol{\Theta}_B$)

13      Initialize operation indices $\boldsymbol{q} \in \mathbb{N}^M$

14      **for** $m = 1$ *to* $M$ **do**

15          **for** $u = 1$ *to* $U$ **do**            // $U \in \mathbb{N}$: Number of possible operations

16              $u' \leftarrow$ UPDATEBESTOPERATION(ADJUSTTREE($\boldsymbol{\Theta}_A[m], u$) $\cdot \boldsymbol{\Theta}_B[m], u$)

17          Append $u'$ to $\boldsymbol{q}$            // $\boldsymbol{q} \in \mathbb{N}^M$: Optimal operations

18      **return** $\boldsymbol{p}, \boldsymbol{q}$

---

**Algorithm 2:** Weight matching for soft tree ensembles

---

1   WEIGHTMATCHING($\boldsymbol{\Theta}_A \in \mathbb{R}^{M \times P}$, $\boldsymbol{\Theta}_B \in \mathbb{R}^{M \times P}$)

2      $\boldsymbol{\Theta}_A, \boldsymbol{\Theta}_B \leftarrow$ WEIGHTING($\boldsymbol{\Theta}_A, \boldsymbol{\Theta}_B$)

3      Initialize similarity matrix for each operation $\boldsymbol{S} \in \mathbb{R}^{U \times M \times M}$

4      **for** $u = 1$ *to* $U$ **do**            // $U \in \mathbb{N}$: Number of possible operations

5          **for** $m_A = 1$ *to* $M$ **do**

6              $\boldsymbol{\theta} \leftarrow$ ADJUSTTREE($\boldsymbol{\Theta}_A[m_A], u$)      // $\boldsymbol{\theta} \in \mathbb{R}^P$: Adjusted tree-wise parameters

7              **for** $m_B = 1$ *to* $M$ **do**

8                  $\boldsymbol{S}[u, m_A, m_B] \leftarrow \boldsymbol{\theta} \cdot \boldsymbol{\Theta}_B[m_B]$

9      $\boldsymbol{S}' \leftarrow \max(\boldsymbol{S}, \text{axis=0})$            // $\boldsymbol{S}' \in \mathbb{R}^{M \times M}$: Similarity matrix between trees

10      $\boldsymbol{p} \leftarrow$ LINEARSUMASSIGNMENT($\boldsymbol{S}'$)            // $\boldsymbol{p} \in \mathbb{N}^M$: Optimal assignments

11      $\boldsymbol{q} \leftarrow \text{argmax}(\boldsymbol{S}, \text{axis=0})[\boldsymbol{p}]$            // $\boldsymbol{q} \in \mathbb{N}^M$: Optimal operations

12      **return** $\boldsymbol{p}, \boldsymbol{q}$

---

**Complexity.** The time complexity of solving the LAP is $\mathcal{O}(M^3)$ using a modified Jonker-Volgenant algorithm without initialization (Crouse, 2016), where $M$ is the number of trees. This process needs to be performed only once in both WM and AM to consider tree permutation invariance. However, the number of additional invariance patterns $U$ scales rapidly as $D$ increases. In a non-oblivious perfect binary tree with depth $D$, there are $2^D - 1$ splitting nodes, resulting in $2^{2^D - 1}$ possible combinations of sign flips, giving total additional invariances pattern $U = 2^{2^D - 1}$. Additionally, in the case of oblivious trees with depth $D$, the number of splitting rules that consider sign flipping is reduced from $2^{2^D - 1}$ to $2^D$ due to the splitting rule sharing at the same depth. Considering the $D!$ distinct splitting order invariance patterns, we have $U = 2^D D!$. Therefore, for large values of $D$, it becomes impractical to conduct an exhaustive search to consider additional invariances.

In Section 3.3, we will discuss methods to eliminate additional invariance by adjusting the tree structure. This enables efficient matching even for deep models. Additionally, in Section 4.2, we will present numerical experiment results and discuss that the practical motivation to apply these algorithms is limited when targeting deep perfect binary trees.

## 3.3 ARCHITECTURE-DEPENDENCY OF THE INVARIANCES

In Section 3.1, we focused on perfect binary trees as they are most commonly used in soft trees (Frosst & Hinton, 2017; Popov et al., 2020; Hazimeh et al., 2020). However, tree architectures can be flexible, and we show that we can specifically design architecture that has neither subtree flip nor splitting order invariances. This allows efficient matching as it is computationally expensive to consider two such invariances.

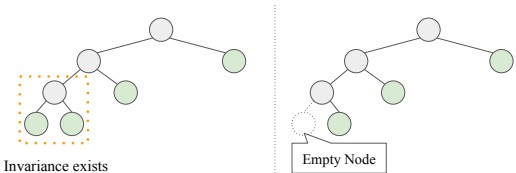

Figure 5: Tree architecture where neither subtree flip invariance nor splitting order invariance exists.

Our idea is to modify a *decision list* shown on the left side of Figure 5, which is a tree structure where branches extend in only one direction. Due to this asymmetric structure, the number of parameters does not increase exponentially with the depth, and the splitting order invariance does not exist. Moreover, subtree flip invariance also does not exist for any internal nodes except for the terminal splitting node, as shown in the left

Table 1: Invariances inherent to each model architecture.

|  | Perm | Flip | Order |
|---|---|---|---|
| Non-Oblivious Tree | ✓ | ✓ | × |
| Oblivious Tree | ✓ | ✓ | ✓ |
| Decision List | ✓ | (✓) | × |
| Decision List (Modified) | ✓ | × | × |

side of Figure 5. To completely remove this invariance, we virtually eliminate one of the terminal leaves by leaving the node empty, that is, a fixed prediction value of zero, as shown on the right side of Figure 5. Therefore only permutation invariance exists for our proposed architecture. We summarize invariances inherent to each model architecture in Table 1.

## 4 EXPERIMENT

We empirically evaluate barriers in soft tree ensembles to examine LMC.

### 4.1 SETUP

**Datasets.** In our experiments, we employed Tabular-Benchmark (Grinsztajn et al., 2022), a collection of tabular datasets suitable for evaluating tree ensembles. Details of datasets are provided in Section A in Appendix. As proposed in Grinsztajn et al. (2022), we randomly sampled $10,000$ instances for train and test data from each dataset. If the dataset contains fewer than $20,000$ instances, they are randomly divided into halves for train and test data. We applied quantile transformation to each feature and standardized it to follow a normal distribution.

**Hyperparameters.** We used three different learning rates $\eta \in \{0.01, 0.001, 0.0001\}$ and adopted the one that yields the highest training accuracy for each dataset. The batch size is set at $512$. It is known that the optimal settings for the learning rate and batch size are interdependent (Smith et al., 2018). Therefore, it is reasonable to fix the batch size while adjusting the learning rate. During AM, we set the amount of data used for random sampling to be the same as the batch size, thus using $512$ samples to measure the similarity of the tree outputs. As the number of trees $M$ and their depths $D$ vary for each experiment, these details will be specified in the experimental results section. During training, we minimized cross-entropy using Adam (Kingma & Ba, 2015) with its default hyperparameters[2]. Training is conducted for $50$ epochs. To measure the barrier using Equation 1, experiments were conducted by interpolating between two models with $\lambda \in \{0, 1/24, \ldots, 23/24, 1\}$, which has the same granularity as in Ainsworth et al. (2023).

**Randomness.** We conducted experiments with five different random seed pairs: $r_A \in \{1, 3, 5, 7, 9\}$ and $r_B \in \{2, 4, 6, 8, 10\}$. As a result, the initial parameters and the contents of the data mini-batches during training are different in each training. In contrast to spawning (Frankle et al., 2020) that branches off from the exact same model partway through, we used more challenging practical conditions. The parameters $w$, $b$, and $\pi$ were randomly initialized using a uniform distribution, identical to the procedure for a fully connected layer in the MLP[3].

---

[2]https://pytorch.org/docs/stable/generated/torch.optim.Adam.html
[3]https://pytorch.org/docs/stable/generated/torch.nn.Linear.html

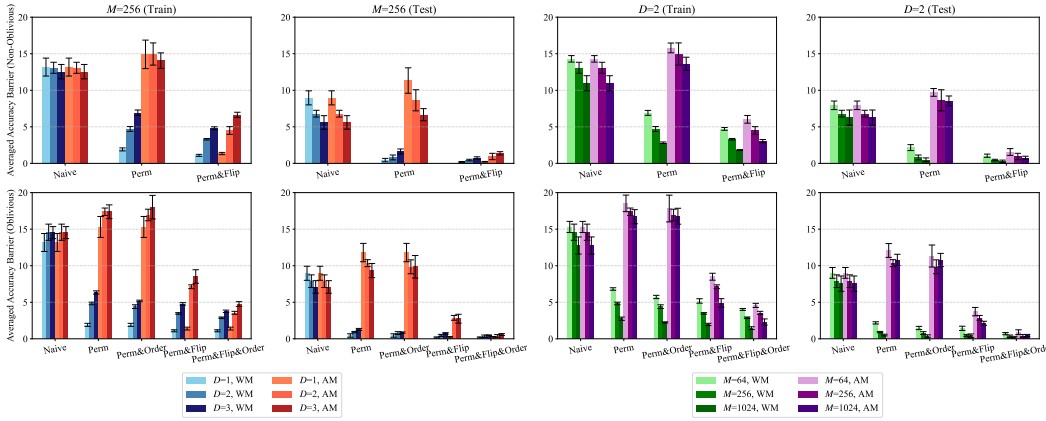

Figure 6: Barriers averaged across 16 datasets with respect to considered invariances for non-oblivious (top row) and oblivious (bottom row) trees. The error bars show the standard deviations of 5 executions.

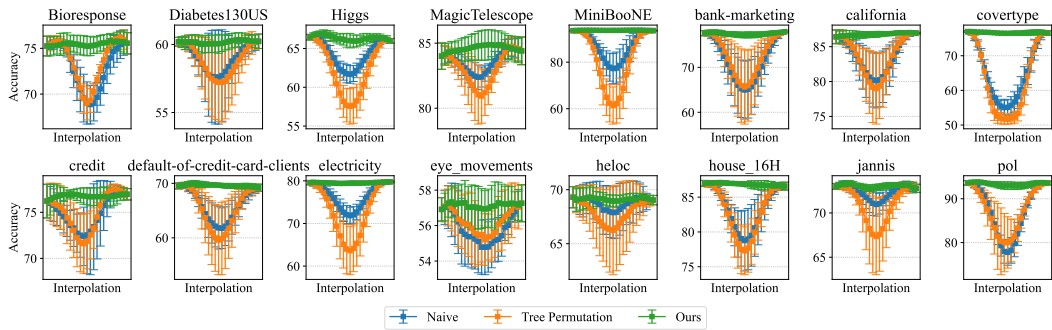

Figure 7: Interpolation curves of test accuracy for oblivious trees on 16 datasets from Tabular-Benchmark (Grinsztajn et al., 2022). Two model pairs are trained on the same dataset. The error bars show the standard deviations of 5 executions.

**Resources.** All experiments were conducted on a system equipped with an Intel Xeon E5-2698 CPU at 2.20 GHz, 252 GB of memory, and Tesla V100-DGXS-32GB GPU, running Ubuntu Linux (version 4.15.0-117-generic). The reproducible PyTorch (Paszke et al., 2019) implementation is provided in the supplementary material.

## 4.2 RESULTS FOR PERFECT BINARY TREES

Figure 6 shows how the barrier between two perfect binary tree model pairs changes in each operation. The vertical axis of each plot in Figure 6 shows the averaged barrier over datasets for each considered invariance. The results for both the oblivious and non-oblivious trees are plotted separately in a vertical layout. The panels on the left display the results when the depth $D$ of the tree varies, keeping $M = 256$ constant. The panels on the right show the results when the number of trees $M$ varies, with $D$ fixed at 2. For both oblivious and non-oblivious trees, we observed that the barrier decreases significantly as the considered invariances increase. Focusing on the test data results, after accounting for various invariances, the barrier is nearly zero, indicating that LMC has been achieved. In particular, the difference between the case of only permutation and the case where additional invariances are considered tends to be larger in the case of AM. This is because parameter values are not used during the rearrangement of the tree in AM. Additionally, it has been observed that the barrier increases as trees become deeper and that the barrier decreases as the number of trees increases. These behaviors correspond to the changes observed in neural networks when the depth varies or when the width of hidden layers increases (Entezari et al., 2022; Ainsworth et al., 2023). Figure 7 shows interpolation curves for AM in oblivious trees with $D = 2$ and $M = 256$. In our figures and tables, "Naive" refers to a straightforward parameter interpolation without any specific optimization; "Tree Permutation"

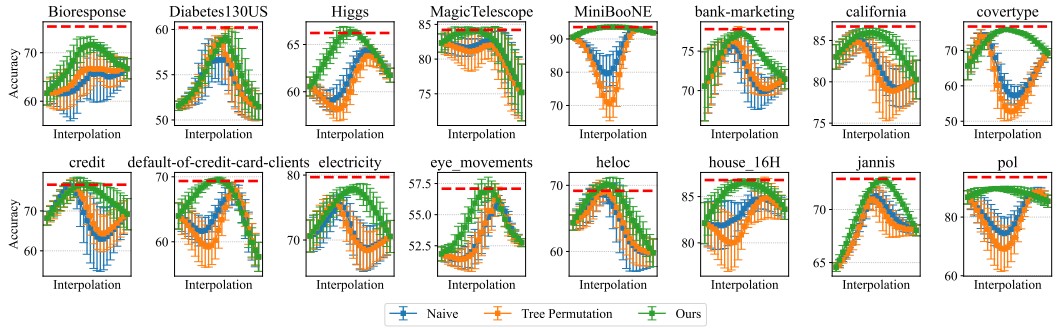

Figure 8: Interpolation curves of test accuracy for oblivious trees on 16 datasets from Tabular-Benchmark (Grinsztajn et al., 2022). Two model pairs are trained on split datasets with different class ratios. The error bars show the standard deviations of 5 executions. Performance of a model trained with the full dataset is shown in the red dashed horizontal lines as a reference.

or "Perm" considers only the permutation invariance; and "Ours" incorporates both the permutation of the trees and tree-inherent invariances. Other details, such as performance for each dataset, are provided in Section B in Appendix.

Furthermore, we conducted experiments with split data following the protocol in Ainsworth et al. (2023) and Jordan et al. (2023), where the initial split consists of randomly sampled 80% negative and 20% positive instances, and the second split inverts these ratios. There is no overlap between the two split datasets. We trained two model pairs using these separately split datasets and observed an improvement in performance by interpolating their parameters. Figure 8 illustrates the interpolation curves under AM in oblivious trees with parameters $D = 2$ and $M = 256$. Through model merging, it demonstrates similar performance to full data training even with split data training for the majority of datasets. Note that the data split is configured to remain consistent even when the training random seeds differ. Detailed results for each dataset using WM or AM are provided in Section B in Appendix.

Table 2 compares the average test barriers of an MLP with a ReLU activation function, whose width is equal to the number of trees, $M = 256$. The procedure for MLPs follows that described in Section 4.1. The permutation for MLPs is optimized using the method described in Ainsworth et al. (2023). Since Ainsworth et al. (2023) indicated that WM outperforms AM in neural networks, WM was used for the comparison. Overall, tree models exhibit smaller barriers compared to MLPs while maintaining similar accuracy levels. It is important to note that MLPs with $D > 1$ tend to have more parameters at the same depth compared to trees, leading to more complex optimization landscapes. Nevertheless, the barrier for the non-oblivious tree at $D = 3$ is smaller than that for the MLP at $D = 2$, even with more parameters. Furthermore, at the same depth of $D = 1$, tree models have a smaller barrier. Here, the model size is evaluated using $F = 44$, the average input feature size of 16 datasets used in the experiments.

Table 2: Barriers, accuracies, and model sizes for MLP, non-oblivious trees, and oblivious trees.

| | | MLP | | |
|---|---|---|---|---|
| **Depth** | **Barrier** | | **Accuracy** | **Size** |
| | **Naive** | **Perm** | | |
| 1 | $8.755 \pm 0.877$ | $\underline{0.491 \pm 0.062}$ | $76.286 \pm 0.094$ | 12034 |
| 2 | $15.341 \pm 1.125$ | $\underline{2.997 \pm 0.709}$ | $75.981 \pm 0.139$ | 77826 |
| 3 | $15.915 \pm 2.479$ | $\underline{5.940 \pm 2.153}$ | $75.935 \pm 0.117$ | 143618 |

| | | Non-Oblivious Tree | | | |
|---|---|---|---|---|---|
| **Depth** | **Barrier** | | | **Accuracy** | **Size** |
| | **Naive** | **Perm** | **Ours** | | |
| 1 | $8.965 \pm 0.963$ | $0.449 \pm 0.235$ | $\underline{0.181 \pm 0.078}$ | $76.464 \pm 0.167$ | 12544 |
| 2 | $6.801 \pm 0.464$ | $0.811 \pm 0.333$ | $\underline{0.455 \pm 0.105}$ | $76.631 \pm 0.052$ | 36608 |
| 3 | $5.602 \pm 0.926$ | $1.635 \pm 0.334$ | $\underline{0.740 \pm 0.158}$ | $76.339 \pm 0.115$ | 84736 |

| | | Oblivious Tree | | | |
|---|---|---|---|---|---|
| **Depth** | **Barrier** | | | **Accuracy** | **Size** |
| | **Naive** | **Perm** | **Ours** | | |
| 1 | $8.965 \pm 0.963$ | $0.449 \pm 0.235$ | $\underline{0.181 \pm 0.078}$ | $76.464 \pm 0.167$ | 12544 |
| 2 | $7.881 \pm 0.866$ | $0.918 \pm 0.092$ | $\underline{0.348 \pm 0.172}$ | $76.623 \pm 0.042$ | 25088 |
| 3 | $7.096 \pm 0.856$ | $1.283 \pm 0.139$ | $\underline{0.484 \pm 0.049}$ | $76.535 \pm 0.063$ | 38656 |

In Section 3.2, we have shown that considering additional invariances for deep perfect binary trees is computationally challenging, which may suggest developing heuristic algorithms for deep trees. However, we consider this to be a low priority, supported by our observations that the barrier tends to increase as trees deepen even if we consider invariances. This trend indicates that deep models are fundamentally less important for model merging considerations. Furthermore, deep perfect binary trees are rarely used in practical scenarios. Kanoh & Sugiyama (2022) demonstrated that generalization performance degrades with increasing depth in perfect binary trees due to the

Table 3: Barriers averaged for 16 datasets under WM with $D = 2$ and $M = 256$.

| Architecture | Train | | | | Test | | | |
| --- | --- | --- | --- | --- | --- | --- | --- | --- |
| | Barrier | | | Accuracy | Barrier | | | Accuracy |
| | Naive | Perm | Ours | | Naive | Perm | Ours | |
| Non-Oblivious Tree | $13.079 \pm 0.755$ | $4.707 \pm 0.332$ | $3.303 \pm 0.104$ | $85.646 \pm 0.090$ | $6.801 \pm 0.464$ | $0.811 \pm 0.333$ | $0.455 \pm 0.105$ | $76.631 \pm 0.052$ |
| Oblivious Tree | $14.580 \pm 1.108$ | $4.834 \pm 0.176$ | $\underline{2.874 \pm 0.108}$ | $85.808 \pm 0.146$ | $7.881 \pm 0.866$ | $0.919 \pm 0.093$ | $\underline{0.348 \pm 0.172}$ | $76.623 \pm 0.042$ |
| Decision List | $13.835 \pm 0.788$ | $3.687 \pm 0.230$ | — | $85.337 \pm 0.134$ | $7.513 \pm 0.944$ | $0.436 \pm 0.120$ | — | $76.629 \pm 0.119$ |
| Decision List (Modified) | $12.922 \pm 1.131$ | $3.328 \pm 0.204$ | — | $85.563 \pm 0.141$ | $6.734 \pm 1.096$ | $0.468 \pm 0.150$ | — | $76.773 \pm 0.051$ |

Table 4: Barriers averaged for 16 datasets under AM with $D = 2$ and $M = 256$.

| Architecture | Train | | | | Test | | | |
| --- | --- | --- | --- | --- | --- | --- | --- | --- |
| | Barrier | | | Accuracy | Barrier | | | Accuracy |
| | Naive | Perm | Ours | | Naive | Perm | Ours | |
| Non-Oblivious Tree | $13.079 \pm 0.755$ | $14.963 \pm 1.520$ | $4.500 \pm 0.527$ | $85.646 \pm 0.090$ | $6.801 \pm 0.464$ | $8.631 \pm 1.444$ | $0.943 \pm 0.435$ | $76.631 \pm 0.052$ |
| Oblivious Tree | $14.580 \pm 1.108$ | $17.380 \pm 0.509$ | $\underline{3.557 \pm 0.201}$ | $85.808 \pm 0.146$ | $7.881 \pm 0.866$ | $10.349 \pm 0.476$ | $\underline{0.395 \pm 0.185}$ | $76.623 \pm 0.042$ |
| Decision List | $13.835 \pm 0.788$ | $12.785 \pm 1.924$ | — | $85.337 \pm 0.134$ | $7.513 \pm 0.944$ | $7.452 \pm 1.840$ | — | $76.629 \pm 0.119$ |
| Decision List (Modified) | $12.922 \pm 1.131$ | $6.364 \pm 0.194$ | — | $85.563 \pm 0.141$ | $6.734 \pm 1.096$ | $2.114 \pm 0.243$ | — | $76.773 \pm 0.051$ |

degeneracy of the Neural Tangent Kernel (NTK) (Jacot et al., 2018). This evidence further supports the preference for shallow perfect binary trees, and increasing the number of trees can enhance the expressive power while reducing barriers.

### 4.3 Results for Decision Lists

We present empirical results of the original decision lists and our modified decision lists, as shown in Figure 5. As we have shown in Table 1, they have fewer invariances.

Figure 9 illustrates barriers as a function of depth, considering only permutation invariance, with $M$ fixed at 256. In this experiment, we have excluded non-oblivious trees from comparison as the number of their parameters exponentially increases as trees deepen, making them infeasible computation. Our proposed modified decision lists reduce the barrier more effectively than both oblivious trees and the original decision lists. However, the barriers of the modified decision lists are still larger than those obtained by considering additional invariances with perfect binary trees. Tables 3 and 4 show the averaged barriers for 16 datasets, with $D = 2$ and $M = 256$. Although the barriers of modified decision lists are small when considering only permutations (Perm), perfect binary trees such as oblivious trees with additional invariances (Ours) exhibit smaller barriers, which supports the validity of using oblivious trees as in Popov et al. (2020) and Chang et al. (2022). To summarize, when considering the practical use of model merging, if the goal is to prioritize efficient computation, we recommend using our proposed decision list. Conversely, if the goal is to prioritize barriers, it would be preferable to use perfect binary trees, which have a greater number of invariances that maintain the functional behavior.

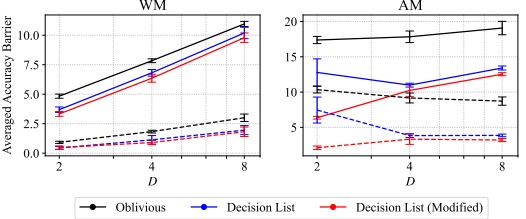

Figure 9: Averaged barrier for 16 datasets as a function of tree depth. The error bars show the standard deviations of 5 executions. The solid line represents the barrier in train accuracy, while the dashed line represents the barrier in test accuracy.

## 5 Conclusion

We have presented the first investigation of LMC for soft tree ensembles. We have identified additional invariances inherent in tree architectures and empirically demonstrated the importance of considering these factors. Achieving LMC is crucial not only for understanding the behavior of non-convex optimization from a learning theory perspective but also for implementing practical techniques such as model merging. By arithmetically combining parameters of differently trained models, a wide range of applications have been explored, such as federated leanning (McMahan et al., 2017) and continual learning (Mirzadeh et al., 2021). Our research extends these techniques to soft tree ensembles. Future work will explore empirical investigations, including the perspective of general mode connectivity (Garipov et al., 2018).

ACKNOWLEDGEMENTS

This work was supported by JSPS, KAKENHI Grant Number JP21H03503, Japan and JST, CREST Grant Number JPMJCR22D3, Japan.

ETHICS STATEMENT

This study provides a fundamental analysis of ensemble learning, and we believe that our discussion will not result in detrimental uses.

REPRODUCIBILITY STATEMENT

Source codes are available in the supplementary material to reproduce the numerical experiments and visualizations.

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

# A DATASET

Details of datasets used in experiments are provided in Table 5.

Table 5: Summary of the datasets used in the experiments.

| Dataset | $N$ | $F$ | Link |
|---|---|---|---|
| Bioresponse | 3434 | 419 | https://www.openml.org/d/45019 |
| Diabetes130US | 71090 | 7 | https://www.openml.org/d/45022 |
| Higgs | 940160 | 24 | https://www.openml.org/d/44129 |
| MagicTelescope | 13376 | 10 | https://www.openml.org/d/44125 |
| MiniBooNE | 72998 | 50 | https://www.openml.org/d/44128 |
| bank-marketing | 10578 | 7 | https://www.openml.org/d/44126 |
| california | 20634 | 8 | https://www.openml.org/d/45028 |
| covertype | 566602 | 10 | https://www.openml.org/d/44121 |
| credit | 16714 | 10 | https://www.openml.org/d/44089 |
| default-of-credit-card-clients | 13272 | 20 | https://www.openml.org/d/45020 |
| electricity | 38474 | 7 | https://www.openml.org/d/44120 |
| eye_movements | 7608 | 20 | https://www.openml.org/d/44130 |
| heloc | 10000 | 22 | https://www.openml.org/d/45026 |
| house_16H | 13488 | 16 | https://www.openml.org/d/44123 |
| jannis | 57580 | 54 | https://www.openml.org/d/45021 |
| pol | 10082 | 26 | https://www.openml.org/d/44122 |

# B ADDITIONAL EMPIRICAL RESULTS

Tables 6, 7, 8 and 9 present the barrier for each dataset with $D = 2$ and $M = 256$. By incorporating additional invariances, it has been possible to consistently reduce the barriers.

Tables 10 and 11 detail the characteristics of the barriers in the decision lists for each dataset with $D = 2$ and $M = 256$. The barriers in the modified decision lists tend to be smaller.

Tables 12 and 13 show the barrier for each model when only considering permutations with $D = 2$ and $M = 256$. It is evident that focusing solely on permutations leads to smaller barriers in the modified decision lists compared to other architectures.

Figures 10, 11, 12, 13, 14, 15, 16 and 17 show the interpolation curves of oblivious trees with $D = 2$ and $M = 256$ across various datasets and configurations. Significant improvements are particularly noticeable in AM, but improvements are also observed in WM. These characteristics are also apparent in the non-oblivious trees, as shown in Figures 18, 19, 20, 21, 22, 23, 24 and 25. Regarding split data training, the dataset for each of the two classes is initially complete (100%). It is then divided into splits of 80% and 20%, and 20% and 80%, respectively. Each model is trained using these splits. Figures 14, 16, 22, and 24 show the training accuracy evaluated using the full dataset (100% for each class). In split data training, the performance reference of full data training is shown only for the performance on the test data. This is because, in split data training, even the training dataset used for evaluation includes portions that are not used for training each model, which differs from the conditions in full data training. In contrast, when evaluating performance on the test data, all of the test data have not been used equally for the training of each model, which allows for a fair comparison between the two approaches. Figures 26, 27, 28, 29, 30, 31, 32, 33, 34, 35, 36, 37, 38, 39, 40 and 41 visualize the same to that of perfect binary trees for the decision lists.

Figures 42 and 43 show the interpolation curves for MNIST (LeCun & Cortes, 2010) with various tree architectures where $D = 2$ and $M = 256$. Although MNIST consists of 2-dimensional image data, it is input as a 1-dimensional vector.

Table 6: Accuracy barrier for non-oblivious trees with WM.

| Dataset | Train | | | Test | | |
|---|---|---|---|---|---|---|
| | Naive | Perm | Perm&Flip | Naive | Perm | Perm&Flip |
| Bioresponse | 18.944 ± 10.076 | 5.876 ± 1.477 | 4.132 ± 0.893 | 8.235 ± 6.456 | 1.285 ± 0.635 | 0.314 ± 0.432 |
| Diabetes130US | 2.148 ± 0.601 | 1.388 ± 1.159 | 0.947 ± 0.888 | 1.014 ± 0.959 | 0.540 ± 0.999 | 0.784 ± 0.840 |
| Higgs | 27.578 ± 1.742 | 18.470 ± 0.769 | 14.772 ± 1.419 | 4.055 ± 1.089 | 0.662 ± 0.590 | 0.292 ± 0.421 |
| MagicTelescope | 2.995 ± 1.198 | 0.576 ± 0.556 | 0.307 ± 0.346 | 2.096 ± 1.055 | 0.361 ± 0.618 | 0.229 ± 0.348 |
| MiniBooNE | 18.238 ± 4.570 | 2.272 ± 0.215 | 1.506 ± 0.211 | 12.592 ± 4.190 | 0.231 ± 0.314 | 0.000 ± 0.000 |
| bank-marketing | 13.999 ± 4.110 | 2.711 ± 1.183 | 1.521 ± 0.463 | 13.593 ± 4.567 | 1.843 ± 1.001 | 0.953 ± 0.688 |
| california | 6.396 ± 2.472 | 0.873 ± 0.551 | 0.520 ± 0.327 | 5.226 ± 2.377 | 0.224 ± 0.248 | 0.206 ± 0.131 |
| covertype | 16.823 ± 4.159 | 1.839 ± 0.336 | 0.914 ± 0.546 | 14.900 ± 4.016 | 1.035 ± 0.106 | 0.376 ± 0.333 |
| credit | 7.317 ± 2.425 | 3.172 ± 2.636 | 2.615 ± 0.831 | 5.861 ± 2.064 | 2.202 ± 3.103 | 1.830 ± 0.588 |
| default-of-credit-card-clients | 14.318 ± 4.509 | 5.419 ± 1.318 | 3.273 ± 0.793 | 6.227 ± 4.205 | 0.937 ± 1.036 | 0.243 ± 0.172 |
| electricity | 10.090 ± 2.930 | 1.035 ± 0.543 | 0.221 ± 0.192 | 9.422 ± 2.795 | 0.771 ± 0.478 | 0.130 ± 0.071 |
| eye_movements | 18.743 ± 1.994 | 11.605 ± 1.927 | 7.866 ± 1.301 | 1.495 ± 0.467 | 0.463 ± 0.183 | 0.180 ± 0.206 |
| heloc | 4.434 ± 1.611 | 1.652 ± 0.475 | 1.012 ± 0.481 | 0.830 ± 0.727 | 0.475 ± 0.447 | 0.322 ± 0.338 |
| house_16H | 8.935 ± 2.504 | 3.362 ± 0.482 | 2.660 ± 1.208 | 4.230 ± 2.189 | 0.219 ± 0.224 | 0.404 ± 0.782 |
| jannis | 17.756 ± 3.322 | 10.442 ± 1.404 | 7.362 ± 0.219 | 3.205 ± 2.849 | 0.029 ± 0.064 | 0.007 ± 0.016 |
| pol | 20.542 ± 2.873 | 4.612 ± 0.912 | 3.225 ± 1.080 | 15.830 ± 2.562 | 1.708 ± 0.599 | 1.012 ± 0.859 |

Table 7: Accuracy barrier for non-oblivious trees with AM.

| Dataset | Train | | | Test | | |
|---|---|---|---|---|---|---|
| | Naive | Perm | Perm&Flip | Naive | Perm | Perm&Flip |
| Bioresponse | 18.944 ± 10.076 | 14.066 ± 7.045 | 5.710 ± 0.915 | 8.235 ± 6.456 | 5.037 ± 3.141 | 0.966 ± 0.316 |
| Diabetes130US | 2.148 ± 0.601 | 3.086 ± 2.566 | 0.574 ± 0.365 | 1.014 ± 0.959 | 1.936 ± 2.878 | 0.105 ± 0.152 |
| Higgs | 27.578 ± 1.742 | 30.704 ± 2.899 | 18.435 ± 1.599 | 4.055 ± 1.089 | 7.272 ± 1.089 | 1.044 ± 0.483 |
| MagicTelescope | 2.995 ± 1.198 | 3.309 ± 1.486 | 0.778 ± 0.515 | 2.096 ± 1.055 | 2.693 ± 1.319 | 0.428 ± 0.327 |
| MiniBooNE | 18.238 ± 4.570 | 34.934 ± 8.157 | 2.332 ± 0.383 | 12.592 ± 4.190 | 28.721 ± 7.869 | 0.074 ± 0.081 |
| bank-marketing | 13.999 ± 4.110 | 13.598 ± 7.638 | 3.098 ± 0.539 | 13.593 ± 4.567 | 12.810 ± 7.605 | 2.643 ± 0.704 |
| california | 6.396 ± 2.472 | 5.800 ± 2.036 | 0.697 ± 0.535 | 5.226 ± 2.377 | 4.858 ± 2.017 | 0.261 ± 0.285 |
| covertype | 16.823 ± 4.159 | 19.708 ± 6.392 | 1.420 ± 0.619 | 14.900 ± 4.016 | 17.765 ± 6.400 | 0.758 ± 0.540 |
| credit | 7.317 ± 2.425 | 10.556 ± 8.753 | 3.640 ± 1.624 | 5.861 ± 2.064 | 9.378 ± 9.083 | 2.551 ± 1.987 |
| default-of-credit-card-clients | 14.318 ± 4.509 | 14.166 ± 2.297 | 4.247 ± 1.678 | 6.227 ± 4.205 | 6.514 ± 2.049 | 0.885 ± 1.852 |
| electricity | 10.090 ± 2.930 | 12.955 ± 4.558 | 0.762 ± 0.332 | 9.422 ± 2.795 | 12.261 ± 4.554 | 0.499 ± 0.260 |
| eye_movements | 18.743 ± 1.994 | 18.757 ± 1.273 | 10.957 ± 1.019 | 1.495 ± 0.467 | 1.583 ± 1.011 | 0.146 ± 0.167 |
| heloc | 4.434 ± 1.611 | 6.564 ± 2.404 | 1.774 ± 0.672 | 0.830 ± 0.727 | 2.179 ± 2.100 | 0.385 ± 0.370 |
| house_16H | 8.935 ± 2.504 | 10.184 ± 2.667 | 3.908 ± 0.863 | 4.230 ± 2.189 | 5.664 ± 2.461 | 1.056 ± 0.693 |
| jannis | 17.756 ± 3.322 | 19.004 ± 1.246 | 9.890 ± 1.036 | 3.205 ± 2.849 | 4.047 ± 1.415 | 0.346 ± 0.443 |
| pol | 20.542 ± 2.873 | 16.267 ± 3.914 | 7.967 ± 3.208 | 15.830 ± 2.562 | 12.863 ± 3.983 | 4.539 ± 2.727 |

Table 8: Accuracy barrier for oblivious trees with WM.

| Dataset | Train | | | Test | | |
|---|---|---|---|---|---|---|
| | Naive | Perm | Perm&Order&Flip | Naive | Perm | Perm&Order&Flip |
| Bioresponse | 16.642 ± 4.362 | 4.800 ± 0.895 | 3.289 ± 0.680 | 7.165 ± 2.547 | 1.069 ± 1.020 | 0.299 ± 0.247 |
| Diabetes130US | 3.170 ± 3.304 | 1.120 ± 1.123 | 0.246 ± 0.177 | 2.831 ± 3.476 | 0.882 ± 1.309 | 0.181 ± 0.155 |
| Higgs | 28.640 ± 0.914 | 19.754 ± 1.023 | 13.689 ± 0.814 | 4.648 ± 0.966 | 1.270 ± 0.808 | 0.266 ± 0.232 |
| MagicTelescope | 2.659 ± 1.637 | 0.473 ± 0.632 | 0.077 ± 0.110 | 2.012 ± 1.343 | 0.534 ± 0.565 | 0.093 ± 0.144 |
| MiniBooNE | 22.344 ± 7.001 | 2.388 ± 0.194 | 1.628 ± 0.208 | 16.454 ± 6.706 | 0.075 ± 0.086 | 0.012 ± 0.019 |
| bank-marketing | 13.512 ± 6.416 | 2.998 ± 1.582 | 0.925 ± 0.688 | 12.856 ± 6.609 | 2.324 ± 1.618 | 0.634 ± 0.433 |
| california | 8.281 ± 4.253 | 0.874 ± 0.524 | 0.351 ± 0.267 | 6.578 ± 4.264 | 0.342 ± 0.209 | 0.034 ± 0.024 |
| covertype | 23.977 ± 2.565 | 2.073 ± 0.657 | 0.976 ± 0.523 | 21.790 ± 2.253 | 0.992 ± 0.496 | 0.422 ± 0.319 |
| credit | 6.912 ± 4.083 | 2.369 ± 0.887 | 0.662 ± 0.606 | 5.739 ± 4.502 | 1.324 ± 0.674 | 0.350 ± 0.522 |
| default-of-credit-card-clients | 16.301 ± 4.462 | 4.512 ± 1.033 | 2.902 ± 0.620 | 7.618 ± 3.873 | 0.728 ± 0.331 | 0.531 ± 0.557 |
| electricity | 8.835 ± 1.824 | 1.060 ± 0.684 | 0.279 ± 0.266 | 7.952 ± 1.995 | 0.731 ± 0.383 | 0.285 ± 0.200 |
| eye_movements | 22.604 ± 1.486 | 12.687 ± 1.645 | 7.826 ± 1.822 | 2.884 ± 1.646 | 0.825 ± 0.711 | 0.607 ± 0.259 |
| heloc | 6.282 ± 2.351 | 2.517 ± 1.156 | 1.507 ± 0.498 | 1.625 ± 1.480 | 0.869 ± 0.957 | 0.727 ± 0.785 |
| house_16H | 13.600 ± 5.135 | 3.302 ± 0.376 | 1.950 ± 0.346 | 8.055 ± 4.429 | 0.330 ± 0.441 | 0.158 ± 0.098 |
| jannis | 19.390 ± 1.013 | 11.358 ± 0.377 | 7.140 ± 0.538 | 1.999 ± 1.237 | 0.305 ± 0.409 | 0.214 ± 0.235 |
| pol | 20.125 ± 2.902 | 5.059 ± 1.482 | 2.544 ± 1.005 | 15.887 ± 3.061 | 2.100 ± 1.358 | 0.751 ± 0.892 |

Table 9: Accuracy barrier for oblivious trees with AM.

| Dataset | Train | | | Test | | |
|---|---|---|---|---|---|---|
| | Naive | Perm | Perm&Order&Flip | Naive | Perm | Perm&Order&Flip |
| Bioresponse | 16.642 ± 4.362 | 19.033 ± 8.533 | 6.358 ± 1.915 | 7.165 ± 2.547 | 6.904 ± 5.380 | 1.038 ± 0.591 |
| Diabetes130US | 3.170 ± 3.304 | 5.473 ± 3.260 | 0.703 ± 0.517 | 2.831 ± 3.476 | 5.290 ± 3.486 | 0.390 ± 0.291 |
| Higgs | 28.640 ± 0.914 | 33.234 ± 3.164 | 15.678 ± 0.713 | 4.648 ± 0.966 | 8.113 ± 2.614 | 0.415 ± 0.454 |
| MagicTelescope | 2.659 ± 1.637 | 3.902 ± 1.931 | 0.224 ± 0.256 | 2.012 ± 1.343 | 3.687 ± 1.876 | 0.334 ± 0.434 |
| MiniBooNE | 22.344 ± 7.001 | 41.022 ± 3.398 | 2.184 ± 0.425 | 16.454 ± 6.706 | 34.452 ± 3.161 | 0.033 ± 0.056 |
| bank-marketing | 13.512 ± 6.416 | 12.248 ± 6.748 | 1.330 ± 0.806 | 12.856 ± 6.609 | 11.356 ± 7.168 | 0.695 ± 0.464 |
| california | 8.281 ± 4.253 | 9.539 ± 4.798 | 0.371 ± 0.365 | 6.578 ± 4.264 | 8.354 ± 4.648 | 0.112 ± 0.181 |
| covertype | 23.977 ± 2.565 | 27.590 ± 2.172 | 1.051 ± 0.407 | 21.790 ± 2.253 | 25.289 ± 1.787 | 0.403 ± 0.236 |
| credit | 6.912 ± 4.083 | 9.839 ± 6.698 | 1.169 ± 0.839 | 5.739 ± 4.502 | 8.291 ± 7.268 | 0.549 ± 0.751 |
| default-of-credit-card-clients | 16.301 ± 4.462 | 21.746 ± 7.075 | 3.646 ± 0.520 | 7.618 ± 3.873 | 12.183 ± 5.954 | 0.285 ± 0.372 |
| electricity | 8.835 ± 1.824 | 18.177 ± 5.979 | 0.472 ± 0.507 | 7.952 ± 1.995 | 17.396 ± 5.809 | 0.405 ± 0.356 |
| eye_movements | 22.604 ± 1.486 | 23.221 ± 3.024 | 8.588 ± 2.248 | 2.884 ± 1.646 | 2.761 ± 1.628 | 0.398 ± 0.435 |
| heloc | 6.282 ± 2.351 | 9.074 ± 3.894 | 2.541 ± 0.471 | 1.625 ± 1.480 | 3.891 ± 2.655 | 0.485 ± 0.397 |
| house_16H | 13.600 ± 5.135 | 17.963 ± 5.099 | 2.841 ± 0.543 | 8.055 ± 4.429 | 12.192 ± 4.635 | 0.292 ± 0.157 |
| jannis | 19.390 ± 1.013 | 22.482 ± 3.113 | 9.570 ± 0.316 | 1.999 ± 1.237 | 4.292 ± 2.509 | 0.069 ± 0.154 |
| pol | 20.125 ± 2.902 | 19.558 ± 5.785 | 3.056 ± 0.510 | 15.887 ± 3.061 | 14.858 ± 5.523 | 0.961 ± 0.722 |

Table 10: Accuracy barrier for decision lists with WM.

| Dataset | Train | | | | Test | | | |
|---|---|---|---|---|---|---|---|---|
| | Naive | Perm | Naive (Modified) | Perm (Modified) | Naive | Perm | Naive (Modified) | Perm (Modified) |
| Bioresponse | 21.323 ± 6.563 | 4.259 ± 0.698 | 14.578 ± 3.930 | 4.641 ± 0.918 | 9.325 ± 3.988 | 0.346 ± 0.277 | 7.346 ± 4.261 | 1.309 ± 0.827 |
| Diabetes130US | 5.182 ± 3.745 | 1.483 ± 1.006 | 2.754 ± 1.098 | 1.088 ± 0.608 | 4.910 ± 4.244 | 1.293 ± 1.332 | 1.476 ± 1.308 | 0.849 ± 0.885 |
| Higgs | 27.778 ± 1.036 | 16.110 ± 0.518 | 28.915 ± 1.314 | 14.071 ± 0.395 | 4.777 ± 0.803 | 0.106 ± 0.203 | 5.136 ± 0.946 | 0.039 ± 0.083 |
| MagicTelescope | 4.855 ± 3.388 | 0.355 ± 0.682 | 5.138 ± 2.655 | 0.182 ± 0.141 | 4.137 ± 3.763 | 0.280 ± 0.519 | 4.534 ± 2.588 | 0.157 ± 0.162 |
| MiniBooNE | 23.059 ± 1.479 | 1.911 ± 0.138 | 14.916 ± 3.616 | 1.580 ± 0.178 | 17.248 ± 1.683 | 0.025 ± 0.036 | 9.340 ± 3.585 | 0.035 ± 0.042 |
| bank-marketing | 11.952 ± 3.794 | 0.979 ± 0.478 | 11.589 ± 2.167 | 0.373 ± 0.448 | 11.387 ± 4.113 | 0.536 ± 0.472 | 10.540 ± 2.067 | 0.349 ± 0.348 |
| california | 6.522 ± 3.195 | 0.621 ± 0.363 | 8.435 ± 3.273 | 0.538 ± 0.214 | 5.167 ± 2.962 | 0.236 ± 0.146 | 6.844 ± 3.087 | 0.151 ± 0.147 |
| covertype | 13.408 ± 3.839 | 1.341 ± 0.433 | 11.114 ± 2.689 | 1.257 ± 0.904 | 11.162 ± 3.620 | 0.472 ± 0.340 | 8.826 ± 2.729 | 0.477 ± 0.889 |
| credit | 11.238 ± 8.115 | 1.968 ± 0.990 | 14.626 ± 5.448 | 1.390 ± 0.423 | 10.880 ± 9.040 | 1.421 ± 1.046 | 13.667 ± 5.951 | 0.940 ± 0.612 |
| default-of-credit-card-clients | 12.513 ± 5.116 | 3.107 ± 1.123 | 11.378 ± 2.123 | 3.793 ± 0.881 | 5.161 ± 4.304 | 0.328 ± 0.512 | 3.197 ± 1.916 | 0.666 ± 0.651 |
| electricity | 6.524 ± 1.863 | 0.725 ± 0.451 | 9.101 ± 2.685 | 0.944 ± 0.557 | 5.834 ± 1.838 | 0.420 ± 0.354 | 8.487 ± 2.460 | 0.543 ± 0.511 |
| eye_movements | 19.125 ± 1.791 | 9.433 ± 1.385 | 19.738 ± 1.490 | 8.755 ± 1.391 | 1.990 ± 1.623 | 0.329 ± 0.102 | 1.916 ± 1.492 | 0.277 ± 0.302 |
| heloc | 4.513 ± 1.826 | 1.564 ± 0.617 | 5.116 ± 0.793 | 1.574 ± 0.154 | 0.725 ± 0.598 | 0.155 ± 0.190 | 1.263 ± 0.711 | 0.359 ± 0.346 |
| house_16H | 9.195 ± 2.408 | 2.520 ± 0.446 | 8.693 ± 1.302 | 2.222 ± 0.730 | 4.629 ± 2.314 | 0.063 ± 0.129 | 4.192 ± 1.517 | 0.185 ± 0.296 |
| jannis | 20.766 ± 2.097 | 9.484 ± 0.371 | 20.520 ± 1.017 | 7.400 ± 0.324 | 3.947 ± 2.605 | 0.006 ± 0.013 | 4.451 ± 1.300 | 0.004 ± 0.009 |
| pol | 23.401 ± 5.448 | 3.137 ± 1.038 | 20.137 ± 4.200 | 3.435 ± 0.675 | 18.933 ± 5.249 | 0.952 ± 0.925 | 16.522 ± 3.502 | 1.143 ± 0.565 |

Table 11: Accuracy barrier for decision lists with AM.

| Dataset | Train | | | | Test | | | |
|---|---|---|---|---|---|---|---|---|
| | Naive | Perm | Naive (Modified) | Perm (Modified) | Naive | Perm | Naive (Modified) | Perm (Modified) |
| Bioresponse | 21.323 ± 6.563 | 13.349 ± 5.943 | 14.578 ± 3.930 | 10.363 ± 7.256 | 9.325 ± 3.988 | 4.817 ± 2.825 | 7.346 ± 4.261 | 3.871 ± 4.608 |
| Diabetes130US | 5.182 ± 3.745 | 5.590 ± 3.328 | 2.754 ± 1.098 | 1.371 ± 0.507 | 4.910 ± 4.244 | 4.926 ± 3.796 | 1.476 ± 1.308 | 0.694 ± 0.649 |
| Higgs | 27.778 ± 1.036 | 28.910 ± 2.132 | 28.915 ± 1.314 | 20.131 ± 1.693 | 4.777 ± 0.803 | 6.722 ± 1.231 | 5.136 ± 0.946 | 1.755 ± 1.403 |
| MagicTelescope | 4.855 ± 3.388 | 3.349 ± 3.273 | 5.138 ± 2.655 | 1.451 ± 0.705 | 4.137 ± 3.763 | 3.001 ± 3.478 | 4.534 ± 2.588 | 1.090 ± 0.437 |
| MiniBooNE | 23.059 ± 1.479 | 18.149 ± 7.500 | 14.916 ± 3.616 | 3.870 ± 1.168 | 17.248 ± 1.683 | 13.868 ± 7.222 | 9.340 ± 3.585 | 0.797 ± 0.860 |
| bank-marketing | 11.952 ± 3.794 | 9.782 ± 6.722 | 11.589 ± 2.167 | 2.815 ± 0.957 | 11.387 ± 4.113 | 9.151 ± 7.204 | 10.540 ± 2.067 | 2.521 ± 1.055 |
| california | 6.522 ± 3.195 | 5.812 ± 2.365 | 8.435 ± 3.273 | 2.254 ± 0.813 | 5.167 ± 2.962 | 4.899 ± 2.018 | 6.844 ± 3.087 | 1.186 ± 0.643 |
| covertype | 13.408 ± 3.839 | 14.727 ± 7.029 | 11.114 ± 2.689 | 4.036 ± 1.450 | 11.162 ± 3.620 | 13.352 ± 7.056 | 8.826 ± 2.729 | 2.656 ± 1.302 |
| credit | 11.238 ± 8.115 | 18.620 ± 9.806 | 14.626 ± 5.448 | 8.979 ± 6.919 | 10.880 ± 9.040 | 18.606 ± 10.015 | 13.667 ± 5.951 | 8.113 ± 6.633 |
| default-of-credit-card-clients | 12.513 ± 5.116 | 12.880 ± 5.070 | 11.378 ± 2.123 | 6.055 ± 1.178 | 5.161 ± 4.304 | 6.465 ± 5.062 | 3.197 ± 1.916 | 0.533 ± 0.239 |
| electricity | 6.524 ± 1.863 | 4.988 ± 2.732 | 9.101 ± 2.685 | 3.041 ± 0.676 | 5.834 ± 1.838 | 4.361 ± 2.532 | 8.487 ± 2.460 | 2.637 ± 0.730 |
| eye_movements | 19.125 ± 1.791 | 18.694 ± 1.774 | 19.738 ± 1.490 | 13.408 ± 1.196 | 1.990 ± 1.623 | 3.046 ± 1.625 | 1.916 ± 1.492 | 1.807 ± 1.312 |
| heloc | 4.513 ± 1.826 | 5.504 ± 1.650 | 5.116 ± 0.793 | 3.287 ± 0.758 | 0.725 ± 0.598 | 1.711 ± 1.278 | 1.263 ± 0.711 | 0.528 ± 0.147 |
| house_16H | 9.195 ± 2.408 | 8.591 ± 3.370 | 8.693 ± 1.302 | 3.937 ± 0.816 | 4.629 ± 2.314 | 4.547 ± 2.726 | 4.192 ± 1.517 | 0.751 ± 0.508 |
| jannis | 20.766 ± 2.097 | 20.768 ± 2.200 | 20.520 ± 1.017 | 12.008 ± 0.892 | 3.947 ± 2.605 | 6.472 ± 2.342 | 4.451 ± 1.300 | 0.106 ± 0.162 |
| pol | 23.401 ± 5.448 | 17.384 ± 6.441 | 20.137 ± 4.200 | 10.339 ± 2.743 | 18.933 ± 5.249 | 13.285 ± 5.863 | 16.522 ± 3.502 | 6.492 ± 2.536 |

Table 12: Training accuracy barrier for permuted models with WM. The numbers in parentheses represent the original accuracy.

| Dataset | Non-Oblivious Tree | Oblivious Tree | Decision List | Decision List (Modified) |
|---|---|---|---|---|
| Bioresponse | 5.876 ± 1.477 (93.005) | 4.800 ± 0.895 (91.753) | 4.259 ± 0.698 (91.771) | 4.641 ± 0.918 (90.489) |
| Diabetes130US | 1.388 ± 1.159 (60.686) | 1.120 ± 1.123 (60.567) | 1.483 ± 1.006 (60.425) | 1.088 ± 0.608 (61.178) |
| Higgs | 18.470 ± 0.769 (97.232) | 19.754 ± 1.023 (97.616) | 16.110 ± 0.518 (95.838) | 14.071 ± 0.395 (95.831) |
| MagicTelescope | 0.576 ± 0.556 (84.963) | 0.473 ± 0.632 (84.460) | 0.355 ± 0.682 (84.999) | 0.182 ± 0.141 (85.411) |
| MiniBooNE | 2.272 ± 0.215 (99.980) | 2.388 ± 0.194 (99.980) | 1.911 ± 0.138 (99.977) | 1.580 ± 0.178 (99.976) |
| bank-marketing | 2.711 ± 1.183 (79.490) | 2.998 ± 1.582 (79.351) | 0.979 ± 0.478 (79.166) | 0.373 ± 0.448 (79.709) |
| california | 0.873 ± 0.551 (87.897) | 0.874 ± 0.524 (87.909) | 0.621 ± 0.363 (88.012) | 0.538 ± 0.214 (88.054) |
| covertype | 1.839 ± 0.336 (79.445) | 2.073 ± 0.657 (79.754) | 1.341 ± 0.433 (79.618) | 1.257 ± 0.904 (79.550) |
| credit | 3.172 ± 2.636 (78.679) | 2.369 ± 0.887 (78.231) | 1.968 ± 0.990 (78.166) | 1.390 ± 0.423 (78.905) |
| default-of-credit-card-clients | 5.419 ± 1.318 (78.017) | 4.512 ± 1.033 (78.657) | 3.107 ± 1.123 (77.315) | 3.793 ± 0.881 (78.308) |
| electricity | 1.035 ± 0.543 (80.375) | 1.060 ± 0.684 (80.861) | 0.725 ± 0.451 (80.396) | 0.944 ± 0.557 (80.651) |
| eye_movements | 11.605 ± 1.927 (81.693) | 12.687 ± 1.645 (83.730) | 9.433 ± 1.385 (81.075) | 8.755 ± 1.391 (81.451) |
| heloc | 1.652 ± 0.475 (77.430) | 2.517 ± 1.156 (78.370) | 1.564 ± 0.617 (77.968) | 1.574 ± 0.154 (78.550) |
| house_16H | 3.362 ± 0.482 (93.093) | 3.302 ± 0.376 (93.351) | 2.520 ± 0.446 (92.783) | 2.222 ± 0.730 (93.058) |
| jannis | 10.442 ± 1.404 (100.000) | 11.358 ± 0.377 (100.000) | 9.484 ± 0.371 (100.000) | 7.400 ± 0.324 (100.000) |
| pol | 4.612 ± 0.912 (98.348) | 5.059 ± 1.482 (98.340) | 3.137 ± 1.038 (97.883) | 3.435 ± 0.675 (97.881) |

Table 13: Training accuracy barrier for permuted models with AM. The numbers in parentheses represent the original accuracy.

| Dataset | Non-Oblivious | Oblivious | Decision List | Decision List (Modified) |
|---|---|---|---|---|
| Bioresponse | 14.066 ± 7.045 (93.005) | 19.033 ± 8.533 (91.753) | 13.349 ± 5.943 (91.771) | 10.363 ± 7.256 (90.489) |
| Diabetes130US | 3.086 ± 2.566 (60.686) | 5.473 ± 3.260 (60.567) | 5.590 ± 3.328 (60.425) | 1.371 ± 0.507 (61.178) |
| Higgs | 30.704 ± 2.899 (97.232) | 33.234 ± 3.164 (97.616) | 28.910 ± 2.132 (95.838) | 20.131 ± 1.693 (95.831) |
| MagicTelescope | 3.309 ± 1.486 (84.963) | 3.902 ± 1.931 (84.460) | 3.349 ± 3.273 (84.999) | 1.451 ± 0.705 (85.411) |
| MiniBooNE | 34.934 ± 8.157 (99.980) | 41.022 ± 3.398 (99.980) | 18.149 ± 7.500 (99.977) | 3.870 ± 1.168 (99.976) |
| bank-marketing | 13.598 ± 7.638 (79.490) | 12.248 ± 6.748 (79.351) | 9.782 ± 6.722 (79.166) | 2.815 ± 0.957 (79.709) |
| california | 5.800 ± 2.036 (87.897) | 9.539 ± 4.798 (87.909) | 5.812 ± 2.365 (88.012) | 2.254 ± 0.813 (88.054) |
| covertype | 19.708 ± 6.392 (79.445) | 27.590 ± 2.172 (79.754) | 14.727 ± 7.029 (79.618) | 4.036 ± 1.450 (79.550) |
| credit | 10.556 ± 8.753 (78.679) | 9.839 ± 6.698 (78.231) | 18.620 ± 9.806 (78.166) | 8.979 ± 6.919 (78.905) |
| default-of-credit-card-clients | 14.166 ± 2.297 (78.017) | 21.746 ± 7.075 (78.657) | 12.880 ± 5.070 (77.315) | 6.055 ± 1.178 (78.308) |
| electricity | 12.955 ± 4.558 (80.375) | 18.177 ± 5.979 (80.861) | 4.988 ± 2.732 (80.396) | 3.041 ± 0.676 (80.651) |
| eye_movements | 18.757 ± 1.273 (81.693) | 23.221 ± 3.024 (83.730) | 18.694 ± 1.774 (81.075) | 13.408 ± 1.196 (81.451) |
| heloc | 6.564 ± 2.404 (77.430) | 9.074 ± 3.894 (78.370) | 5.504 ± 1.650 (77.968) | 3.287 ± 0.758 (78.550) |
| house_16H | 10.184 ± 2.667 (93.093) | 17.963 ± 5.099 (93.351) | 8.591 ± 3.370 (92.783) | 3.937 ± 0.816 (93.058) |
| jannis | 19.004 ± 1.246 (100.000) | 22.482 ± 3.113 (100.000) | 20.768 ± 2.200 (100.000) | 12.008 ± 0.892 (100.000) |
| pol | 16.267 ± 3.914 (98.348) | 19.558 ± 5.785 (98.340) | 17.384 ± 6.441 (97.883) | 10.339 ± 2.743 (97.881) |

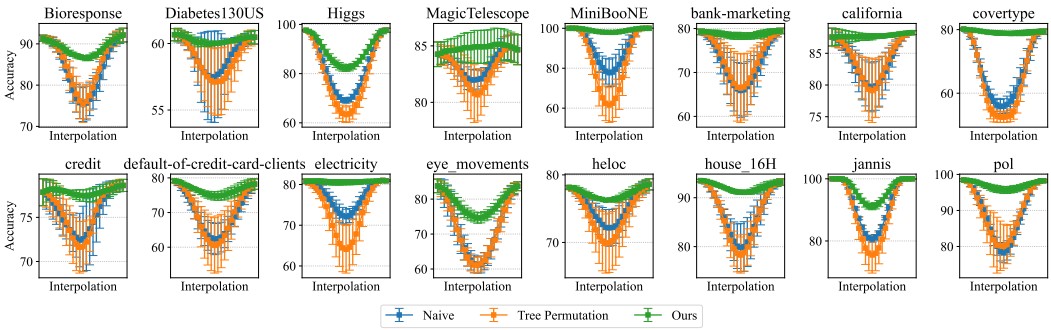

Figure 10: Interpolation curves of train accuracy for oblivious trees with AM.

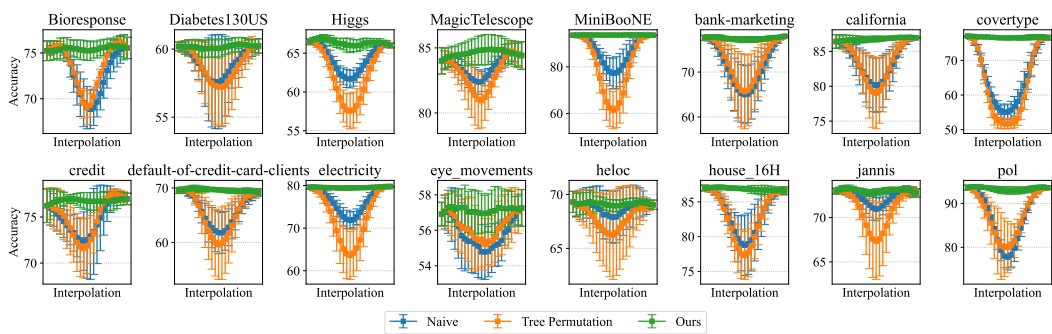

Figure 11: Interpolation curves of test accuracy for oblivious trees with AM.

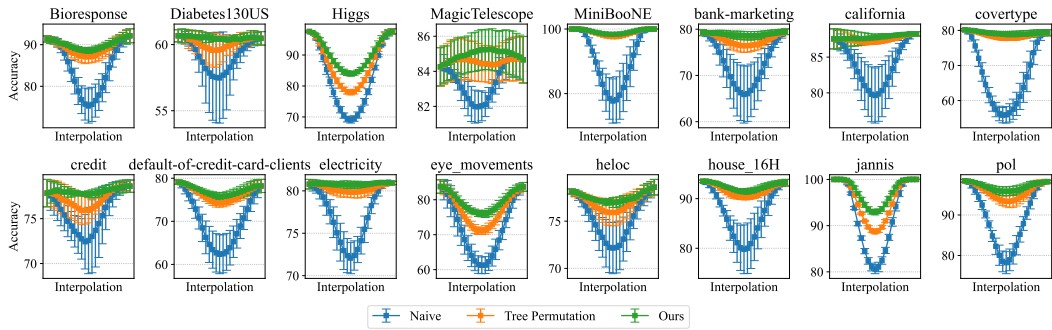

Figure 12: Interpolation curves of train accuracy for oblivious trees with WM.

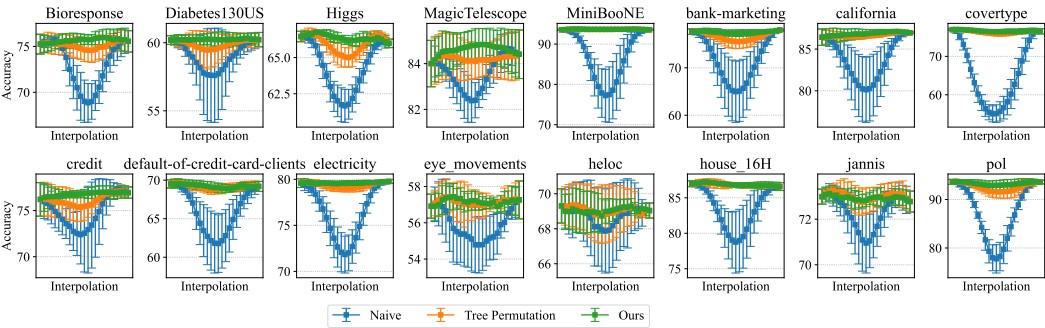

Figure 13: Interpolation curves of test accuracy for oblivious trees with WM.

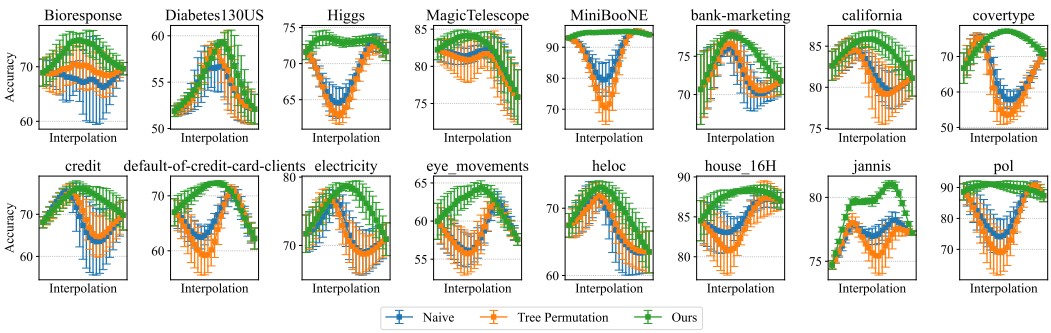

Figure 14: Interpolation curves of train accuracy for oblivious trees with AM by use of split dataset.

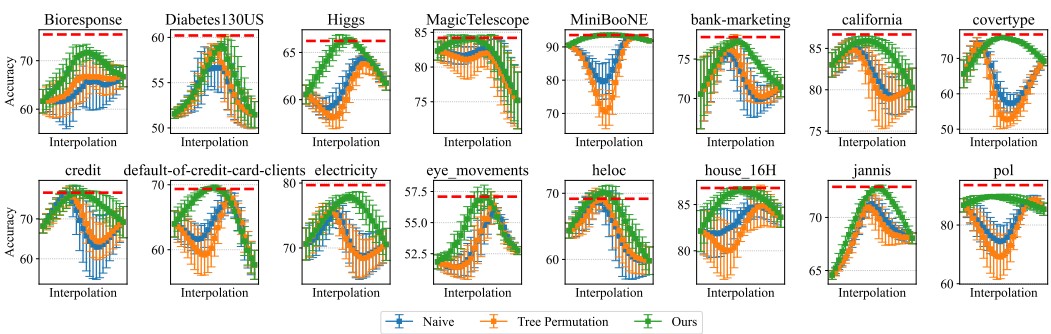

Figure 15: Interpolation curves of test accuracy for oblivious trees with AM by use of split dataset.

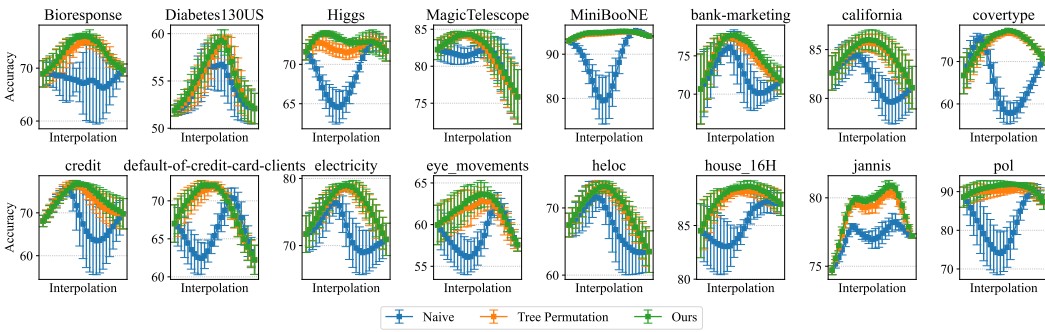

Figure 16: Interpolation curves of train accuracy for oblivious trees with WM by use of split dataset.

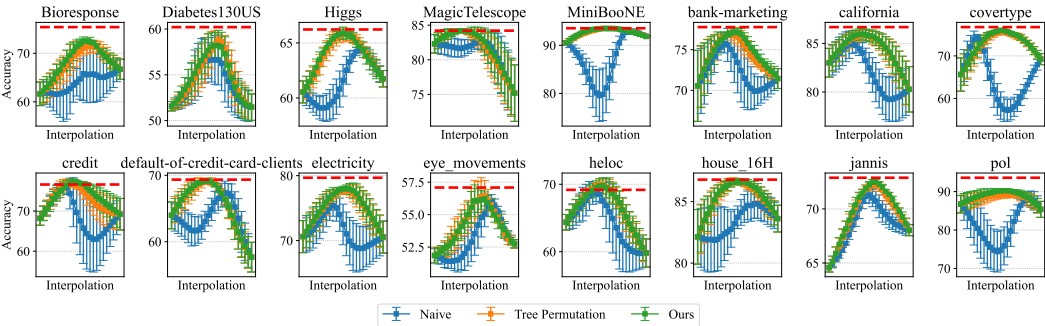

Figure 17: Interpolation curves of test accuracy for oblivious trees with WM by use of split dataset.

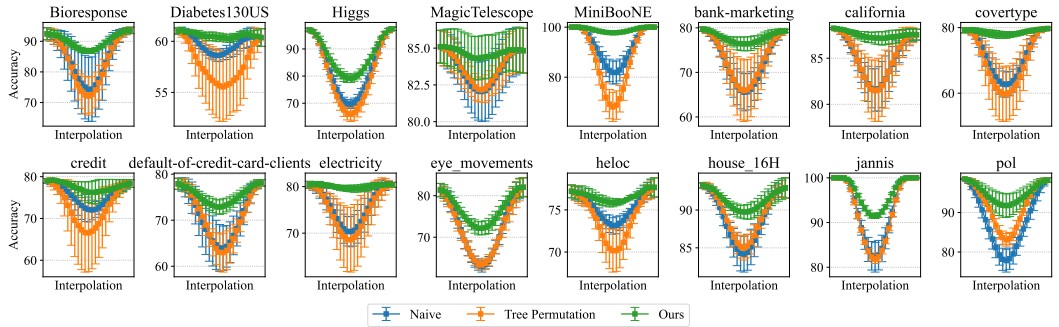

Figure 18: Interpolation curves of train accuracy for non-oblivious trees with AM.

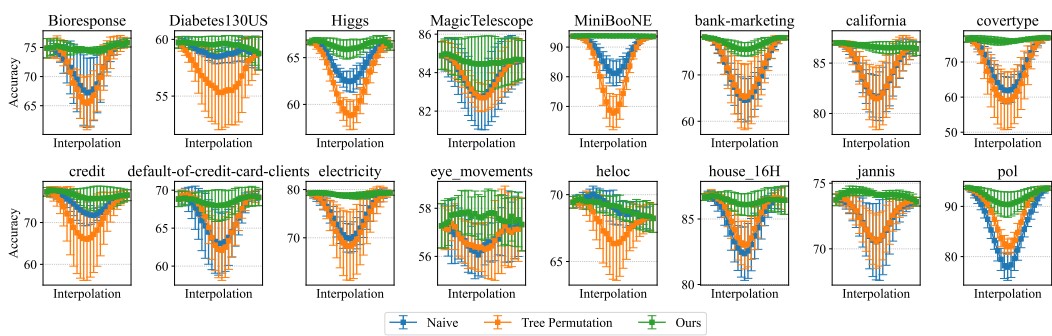

Figure 19: Interpolation curves of test accuracy for non-oblivious trees with AM.

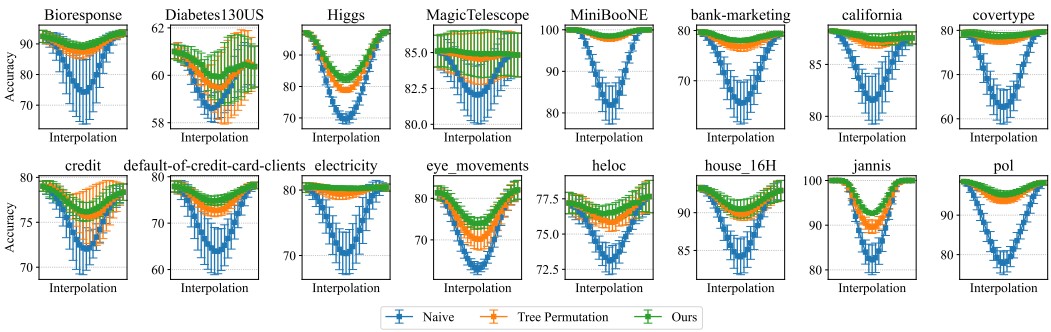

Figure 20: Interpolation curves of train accuracy for non-oblivious trees with WM.

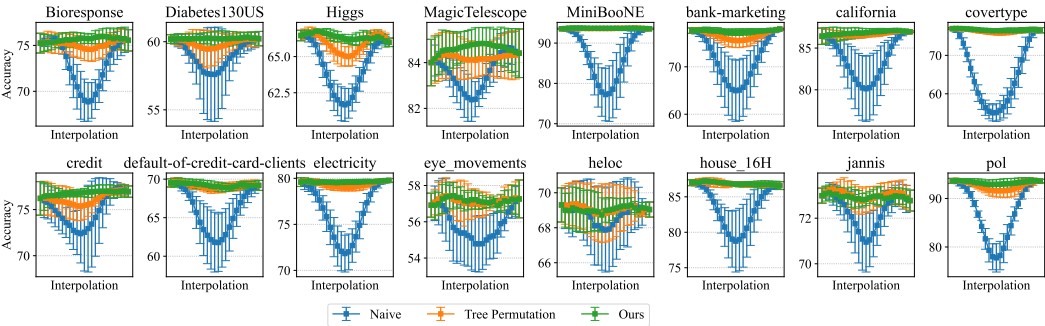

Figure 21: Interpolation curves of test accuracy for non-oblivious trees with WM.

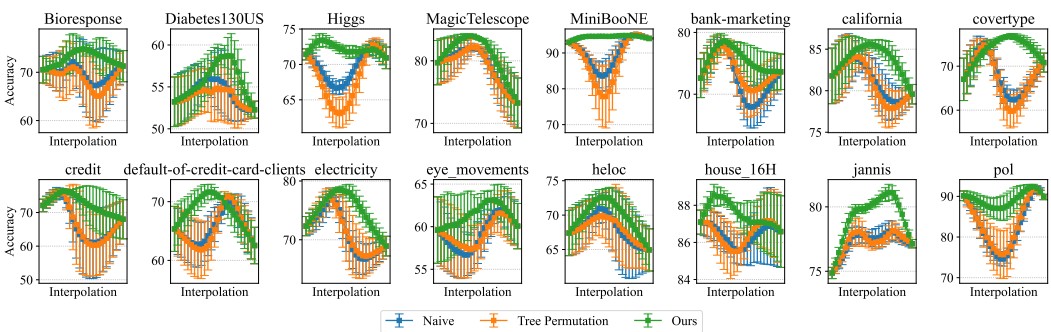

Figure 22: Interpolation curves of train accuracy for non-oblivious trees with AM by use of split dataset.

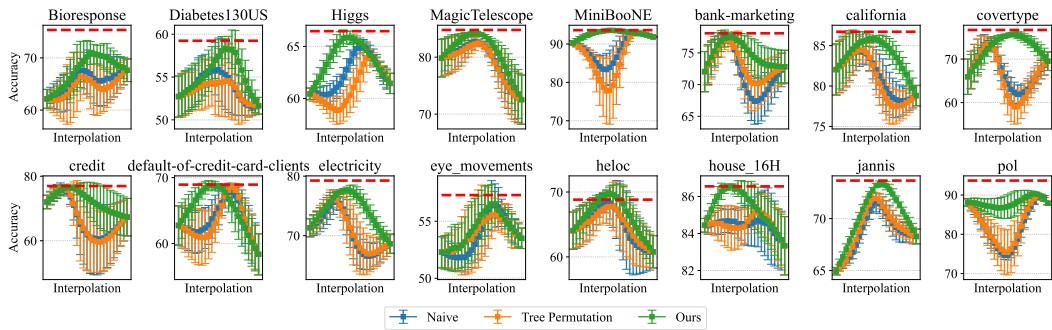

Figure 23: Interpolation curves of test accuracy for non-oblivious trees with AM by use of split dataset.

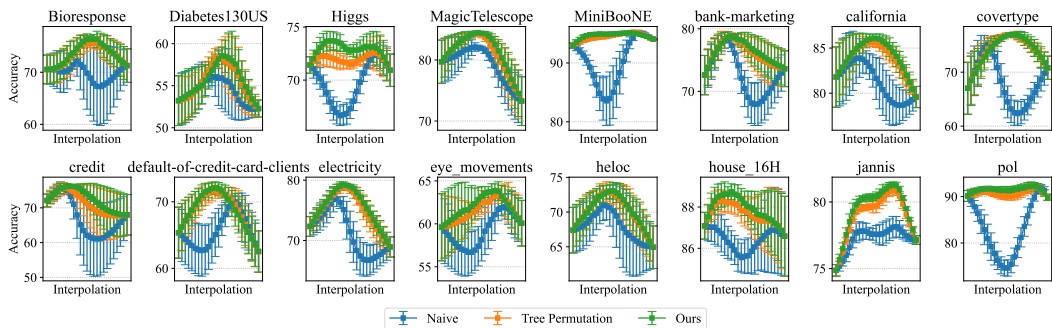

Figure 24: Interpolation curves of train accuracy for non-oblivious trees with WM by use of split dataset.

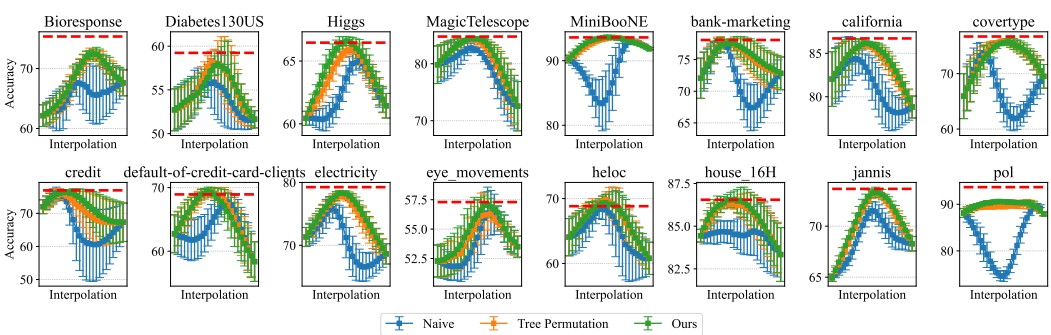

Figure 25: Interpolation curves of test accuracy for non-oblivious trees with WM by use of split dataset.

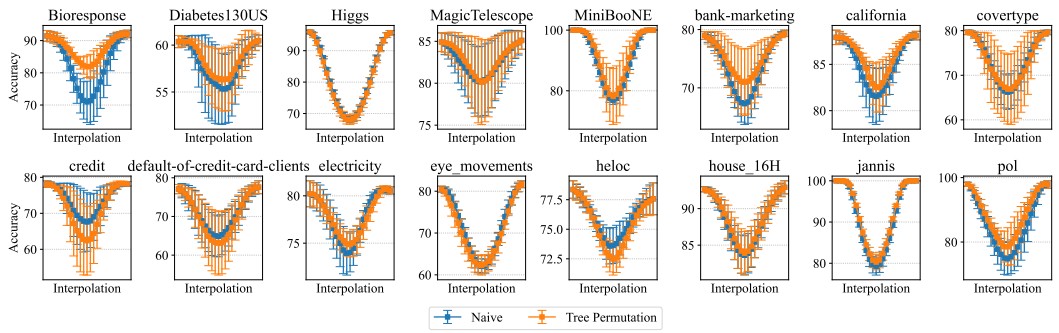

Figure 26: Interpolation curves of train accuracy for decision lists with AM.

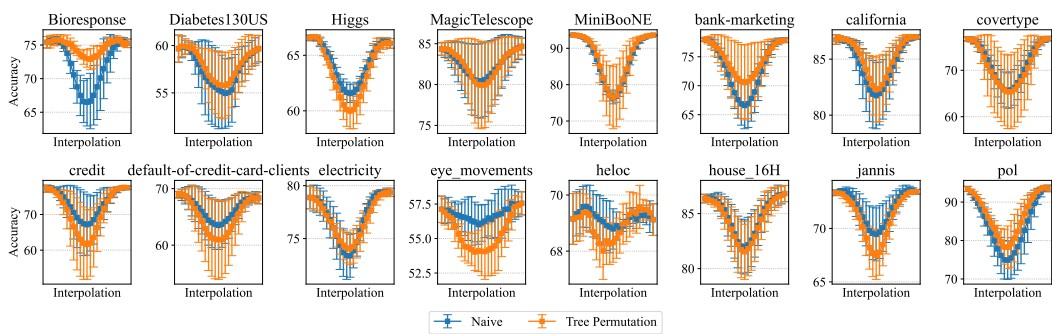

Figure 27: Interpolation curves of test accuracy for decision lists with AM.

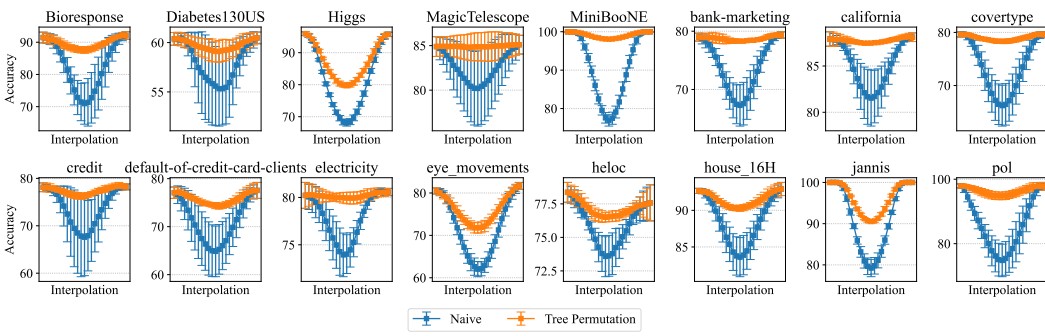

Figure 28: Interpolation curves of train accuracy for decision lists with WM.

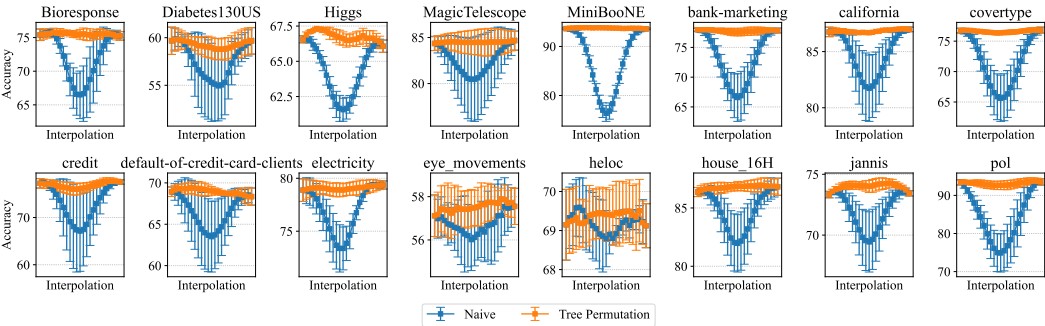

Figure 29: Interpolation curves of test accuracy for decision lists with WM.

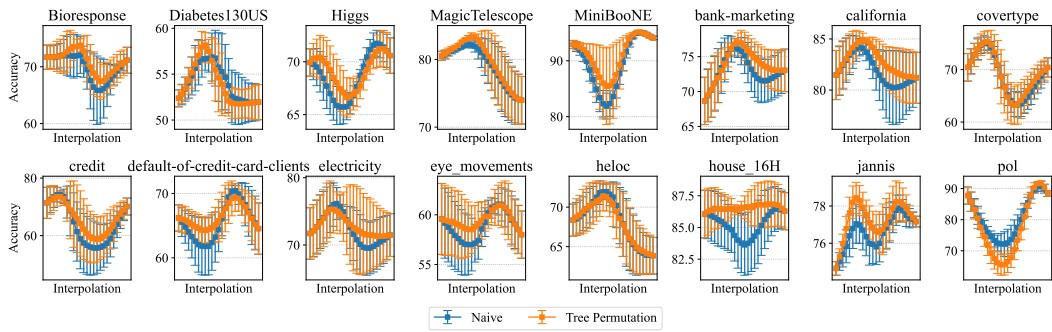

Figure 30: Interpolation curves of train accuracy for decision lists with AM by use of split dataset.

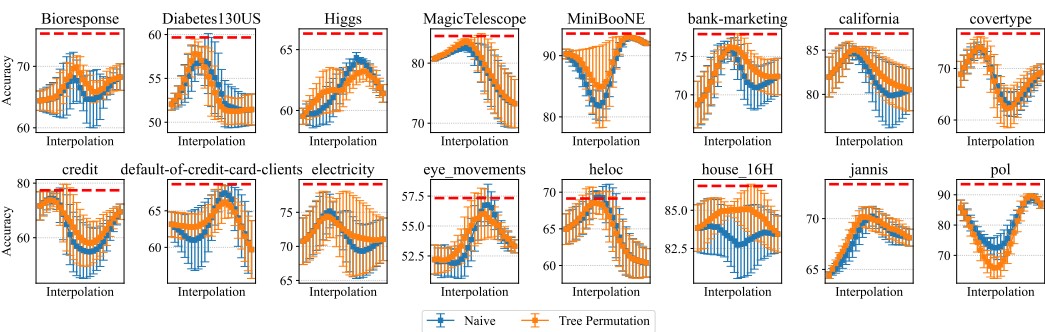

Figure 31: Interpolation curves of test accuracy for decision lists with AM by use of split dataset.

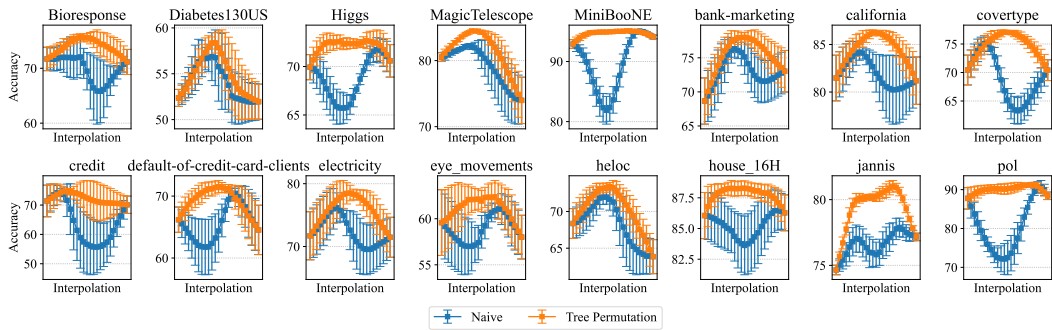

Figure 32: Interpolation curves of train accuracy for decision lists with WM by use of split dataset.

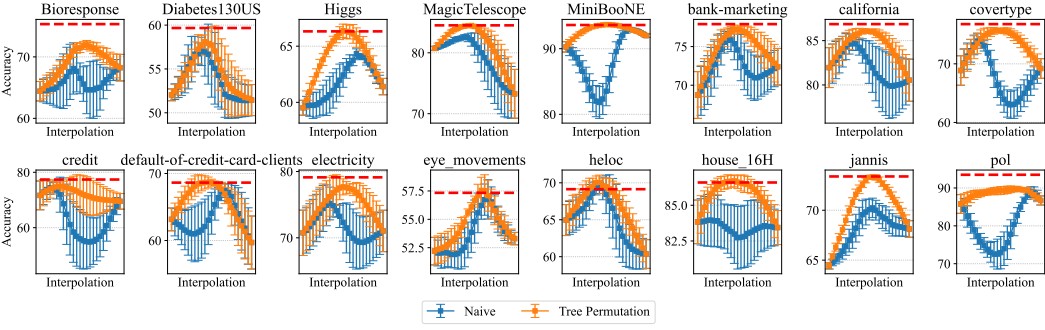

Figure 33: Interpolation curves of test accuracy for decision lists with WM by use of split dataset.

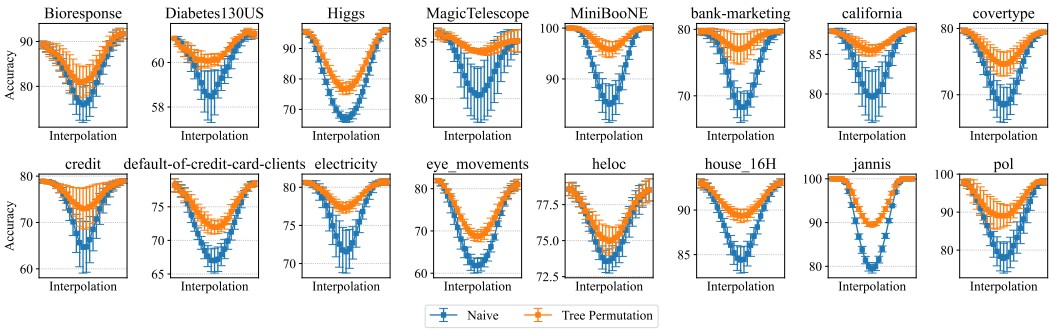

Figure 34: Interpolation curves of train accuracy for modified decision lists with AM.

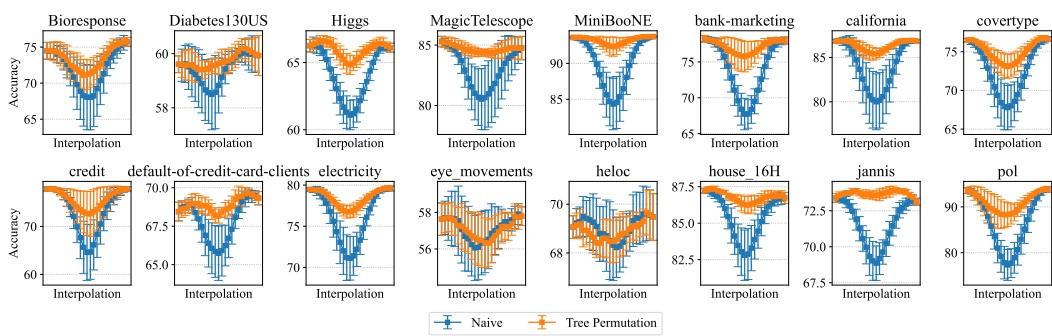

Figure 35: Interpolation curves of test accuracy for modified decision lists with AM.

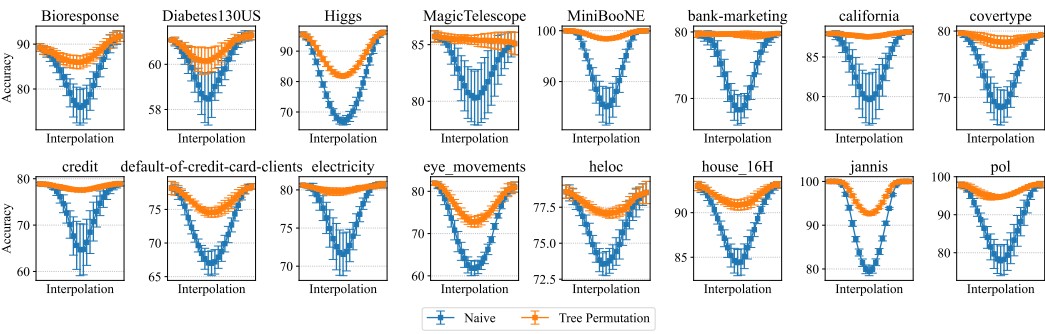

Figure 36: Interpolation curves of train accuracy for modified decision lists with WM.

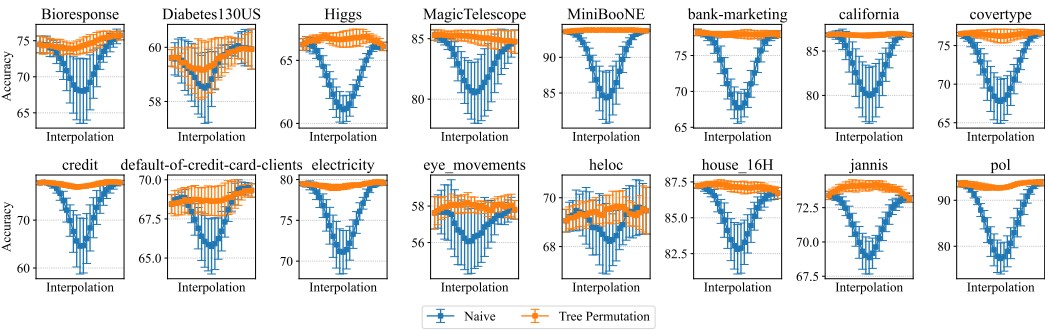

Figure 37: Interpolation curves of test accuracy for modified decision lists with WM.

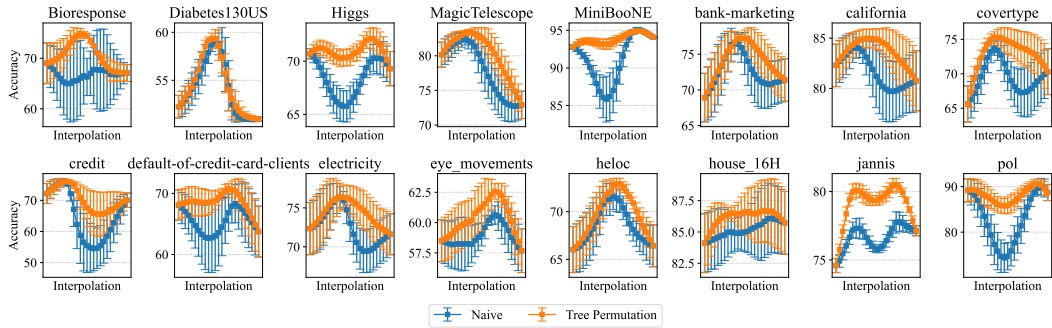

Figure 38: Interpolation curves of train accuracy for modified decision lists with AM by use of split dataset.

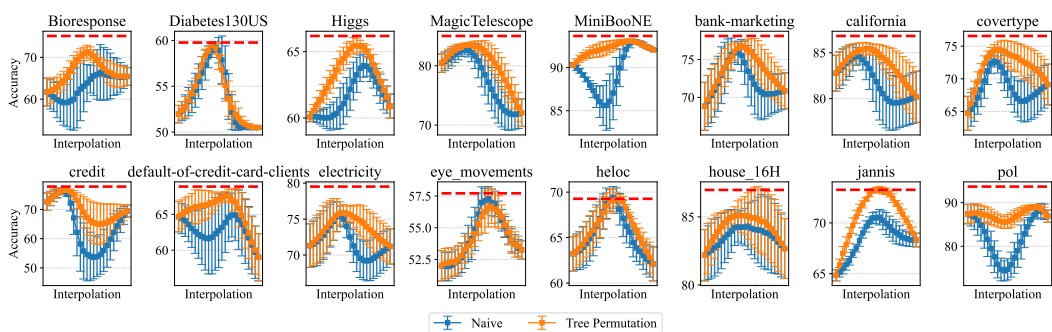

Figure 39: Interpolation curves of test accuracy for modified decision lists with AM by use of split dataset.

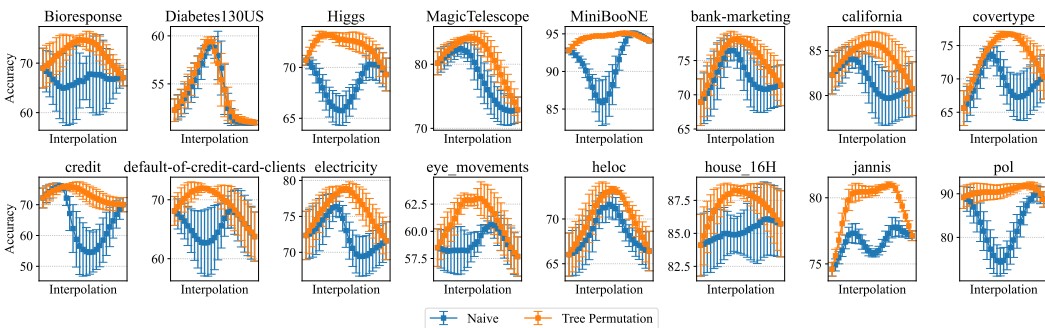

Figure 40: Interpolation curves of train accuracy for modified decision lists with WM by use of split dataset.

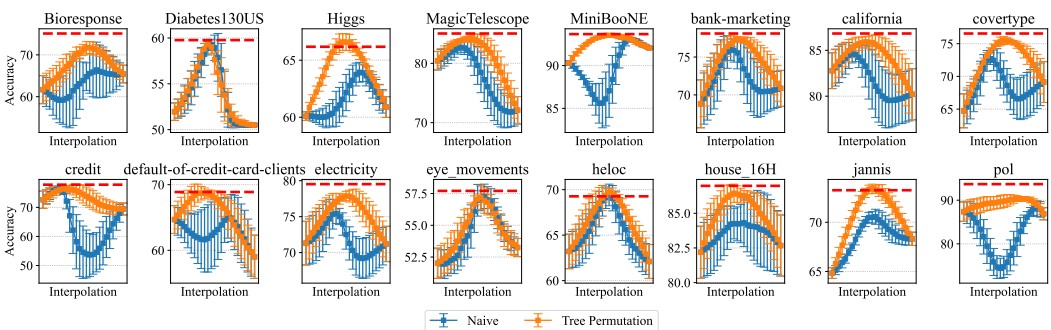

Figure 41: Interpolation curves of test accuracy for modified decision lists with WM by use of split dataset.

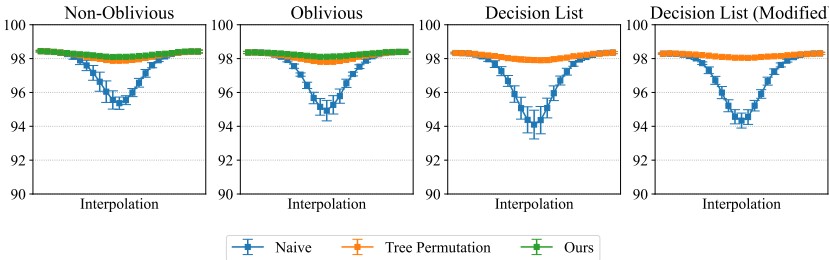

Figure 42: Interpolation curves of test accuracy with WM for MNIST (LeCun & Cortes, 2010) dataset.

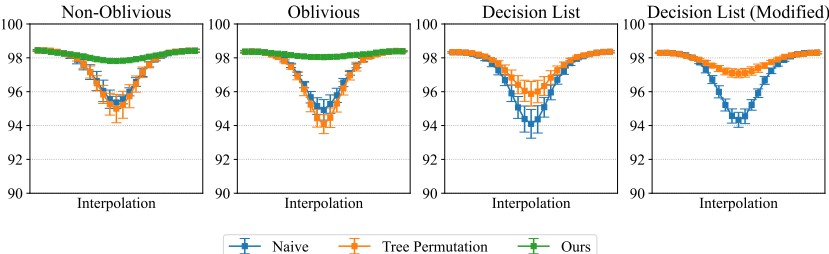

Figure 43: Interpolation curves of test accuracy with AM for MNIST (LeCun & Cortes, 2010) dataset.

