# OpenReview forum: "Linear Mode Connectivity in Differentiable Tree Ensembles"
_ICLR.cc/2025/Conference — ICLR 2025 Spotlight_

### Official Review · Reviewer_rPCT · 2024-11-04

**Soundness:** 3
**Presentation:** 3
**Contribution:** 3
**Rating:** 8
**Confidence:** 2

**Summary:**

This paper investigates Linear Mode Connectivity (LMC) in the context of differentiable tree ensemble models, particularly focusing on soft tree ensembles. LMC refers to the property where linearly interpolating two models (in parameter space) does not degrade performance, a phenomenon previously established for neural networks. The authors extend this concept to tree-based models, identifying specific architectural invariances unique to trees—namely, subtree flip invariance and splitting order invariance—as critical for achieving LMC. They further propose a novel decision list-based architecture that bypasses these invariances, simplifying the conditions necessary for LMC. Through empirical evaluation, the paper demonstrates that LMC can be achieved in tree ensembles, broadening the understanding of LMC beyond neural networks. In summary, I believe that both the research topic and methods are relatively novel and will significantly contribute to the deep learning community.

**Strengths:**

**Originality:** This work is original in extending the concept of LMC to tree-based models, an area previously unexplored. By identifying invariances specific to tree structures, the paper introduces a new perspective on achieving functional equivalence in non-neural models.

**Clarity:** Despite the complexity of the topic, the paper is clearly written, with detailed explanations of both the mathematical framework and empirical results. Figures and diagrams (such as Figure 3 on invariances) effectively support comprehension.

**Significance:** The work expands the applicability of LMC, which is valuable for understanding stability in non-convex optimization and facilitating practical applications such as model merging. The findings on tree-specific invariances could have a substantial impact on future research in tree ensemble models and their integration with neural network techniques.

**Weaknesses:**

**Empirical Validation Scope:** The empirical results are primarily based on tabular datasets. Extending the validation to additional types of data (e.g., image or sequential data processed via tree ensembles) could further strengthen the findings.

**Scalability of Additional Invariance Handling:** The paper discusses the computational expense associated with handling additional invariances for deeper trees. However, a more detailed discussion of potential optimizations or heuristic approaches to manage this complexity would be beneficial.

**Comparative Analysis:** While the results demonstrate LMC in tree ensembles, the paper could benefit from a more in-depth comparison with related methods or model architectures, particularly alternative approaches to model merging or parameter interpolation that do not rely on LMC.

**Equation Reference:** Please use "Equation (1)" instead of "Equation 1."

**Questions:**

See Weakness.

---

> ### Author Response · Authors · 2024-11-21
>
> Thank you for your review.
>
> > Empirical Validation Scope: The empirical results are primarily based on tabular datasets. Extending the validation to additional types of data (e.g., image or sequential data processed via tree ensembles) could further strengthen the findings.
>
> We have additionally conducted experiments using the MNIST dataset, confirming that LMC can be achieved when the image data is flattened into a one-dimensional input. While this is a single instance of other types of data, it provides a foundational proof of concept. Results from the MNIST experiment are included in the Appendix of our revised paper. Since soft tree ensembles are primarily designed as an alternative to MLPs, they are not inherently suited for direct processing of images or sequential data. Therefore our main experiments focus on tabular datasets— the primary domain for tree ensembles. Although it is technically possible to create an end-to-end model by pairing a CNN or RNN module to extract features as a one-dimensional vector, including CNNs or RNNs would complicate the discussion. Therefore, we opted for a simpler experimental setup in this study.
>
> > Scalability of Additional Invariance Handling: The paper discusses the computational expense associated with handling additional invariances for deeper trees. However, a more detailed discussion of potential optimizations or heuristic approaches to manage this complexity would be beneficial.
>
> As discussed in the final paragraph of Section 4.2, the importance of considering additional invariants with an efficient (potentially heuristic) matching algorithm is relatively low. There, we have noted that deep perfect binary trees are not particularly significant, which is supported both empirically and theoretically by the NTK theory [1], and such an efficient matching algorithm would only be necessary for handling those deep perfect binary trees. We have also presented a modified decision list as one of possible solutions to the problem.
> When exact solutions are not required and heuristic methods are acceptable, one possible approach for subtree flip invariance, for example, could involve using a greedy algorithm. For example, to consider subtree flip invariance, this algorithm would sequentially determine the orientation of inequalities, starting from the root of the tree.
>
> > Comparative Analysis: While the results demonstrate LMC in tree ensembles, the paper could benefit from a more in-depth comparison with related methods or model architectures, particularly alternative approaches to model merging or parameter interpolation that do not rely on LMC.
>
> Thank you for your comment. It is important to note that model merging (i.e., linear parameter interpolation) without explicitly assuming LMC often relies on the condition that all training processes start from the same initial parameters. For example, in techniques like Model Soup [2], parameters are merged simply by averaging, without explicitly addressing LMC. However, this approach works because all training starts from the same pretrained model checkpoint, inherently (and implicitly) satisfying the LMC condition. In contrast, our study explores the merging of two models trained from different initial parameters, offering a more generalized framework.
>
> > Equation Reference: Please use "Equation (1)" instead of "Equation 1."
>
> Thank you for your feedback. For ICLR, the LaTeX template is provided on the Author Instruction page, which includes a math reference module (`math_commands.tex`). By using this module, the formatting is naturally set to omit parentheses around equation references.
>
> ```
> % Reference to an equation, upper case
> \def\Eqref#1{Equation~\ref{#1}}
> ```
>
> Currently, we are following this convention. We would appreciate keeping it as is to contribute to a consistent format across the conference.
>
> ----
>
> [1] Kanoh & Sugiyama, A Neural Tangent Kernel Perspective of Infinite Tree Ensembles, ICLR2022
>
> [2] Wortsman et al., Model soups: averaging weights of multiple fine-tuned models improves accuracy without increasing inference time, ICML2022

---

### Official Review · Reviewer_rJGW · 2024-11-04

**Soundness:** 3
**Presentation:** 4
**Contribution:** 2
**Rating:** 6
**Confidence:** 2

**Summary:**

This work studies the phenomenon of linear mode connectivity (LMC) in decision tree models (as opposed to neural networks, where they have been studied extensively). Specifically, they investigate whether soft-tree ensembles can be interpolated, while retaining accuracy. Unlike in neural networks, where permutation invariance is sufficient, the authors show that additional invariances unique to decision tree models are necessary for LMC to hold - in fact, this is the key finding of the work.

**Strengths:**

The paper has several strengths:

- The problem addressed in this work is of general interest, and illuminates an interesting property of decision trees.
- The paper is very clearly written, and easy to follow
- This paper hinges upon it's empirical evaluation: the experiments are quite thorough, and clearly show how incorporating *additional* invariances into model interpolation leads to better accuracy (the key claim of the work).
- The notion of the *modified decision list* architectures is an interesting and potentially highly useful one (as it seems to reduce computation cost per inference)

**Weaknesses:**

There is one significant weakness in this work:

- Lack of Rigorous Theory: the paper would have been strengthened significantly had the authors provided a rigorous analysis of the effect of the stated invariances (permutation, subtree flip, and splitting order) on the ability to interpolate 2 (or more) models - even if only on extremely simple cases. Were that so, the empirical results would have been used to support rigorous theory, which would have been a much more powerful argument.
-  Linear Mode Connectivity was proposed to understand how deep neural networks could be sparsified (Frankle et al, 2020). Can a similar analysis of methods to sparsify decision trees be done, provided the insights proposed in this work? An analysis of this sort - where the need to interpolate models becomes evident - would have also noticeably improved this work.
- The authors do not highlight specific tasks for which interpolation of trees would be of significant benefit. This weakens the argument motivating the need for this work. For instance, does model interpolation in tree ensembles improve robustness to distribution shift?

**Questions:**

See the Weaknesses section.

---

> ### Author Response · Authors · 2024-11-21
>
> Thank you for your review.
>
> > Lack of Rigorous Theory: the paper would have been strengthened significantly had the authors provided a rigorous analysis of the effect of the stated invariances (permutation, subtree flip, and splitting order) on the ability to interpolate 2 (or more) models - even if only on extremely simple cases. Were that so, the empirical results would have been used to support rigorous theory, which would have been a much more powerful argument.
>
> We agree that providing such a theory would be ideal. However, as mentioned in the Introduction, even in the field of neural networks, where research is far more advanced, a theoretical understanding remains an open problem. We believe that our contribution, which examines LMC from a different perspective beyond neural networks, contributes to a deeper understanding of LMC.
>
> > Linear Mode Connectivity was proposed to understand how deep neural networks could be sparsified (Frankle et al, 2020). Can a similar analysis of methods to sparsify decision trees be done, provided the insights proposed in this work? An analysis of this sort - where the need to interpolate models becomes evident - would have also noticeably improved this work.
>
> Yes, the Lottery Ticket Hypothesis (LTH) is closely linked to the concept of LMC as explored in prior research [1] and we believe our work could further connect to these perspectives by expanding on this line of research. The LTH within the context of soft tree ensembles remains largely unexplored, presenting a promising avenue for future study. However, it is important to note that our research focuses on examining LMC between models with different random initializations and different random SGD noise. This setup differs from [1], which analyzes connectivity between two models trained from the same parameters but with different SGD noise, offering a more constrained framework. Our primary motivation is to deepen the understanding of non-convex optimization and explore techniques like model merging. In this regard, our study aligns more closely with [2].
>
> > The authors do not highlight specific tasks for which interpolation of trees would be of significant benefit. This weakens the argument motivating the need for this work. For instance, does model interpolation in tree ensembles improve robustness to distribution shift?
>
> We believe that soft tree ensembles will become a practical choice for a wide range of real-world applications, such as federated learning and continual learning, as highlighted in the conclusion of our revised paper. Soft tree ensembles and neural networks such as MLPs are already being used in practice as one of the options for machine learning models. Our research offers a broader pool of models for trial and error in the context of model merge, which we regard as a valuable contribution. Furthermore, we believe that assessing the robustness of model merging under distribution shifts is a significant topic in its own right. Expanding comparisons across different model types, such as decision trees and neural networks, would further broaden the scope of this research. These considerations represent an important avenue for future work.
>
> ----
>
> [1] Frankle et al., Linear Mode Connectivity and the Lottery Ticket Hypothesis, ICML2020
>
> [2] Ainsworth et al., Git Re-Basin: Merging Models modulo Permutation Symmetries, ICLR2023

---

> > ### Comment · Reviewer_rJGW · 2024-11-25
> > **Response to Rebuttal**
> >
> > Thank you to the authors for the detailed reply. I'm happy to keep my positive appraisal of this work.

---

### Official Review · Reviewer_y2Hs · 2024-11-05

**Soundness:** 3
**Presentation:** 4
**Contribution:** 3
**Rating:** 8
**Confidence:** 2

**Summary:**

This paper explores Linear Mode Connectivity (LMC) in soft tree ensemble models. LMC is well studied in neural networks, but achieving it in tree models presents unique challenges due to their architectural differences. The authors identify additional invariants, subtree flip and splitting order, that are necessary to achieve LMC in tree structures that differ from neural networks, including permutation invariance. With a matching algorithm that takes these invariants into consideration, the authors show that LMC can also be achieved in soft tree ensembles. And they further present a modified decision list architecture that preserves LMC without these additional invariances, aiming to simplify model merging and other parameter-based applications.

**Strengths:**

- The approach of exploring linear mode connectivity, which has been mainly studied in neural networks, in a different architecture is novel.
- The organization and flow of the paper effectively explains the existing research and the proposed method and is easy to understand.
- Through various experiments, the authors show that linear mode connectivity can be achieved in soft tree ensembles.
- The authors show that soft tree ensembles trained on different data splits can be merged to form better performing models.

**Weaknesses:**

- The authors do not provide a more efficient matching algorithm other than exhaustive search.
- The paper does not provide sufficient justification for using the proposed modified decision list. Specifically, the proposed architecture presents a higher barrier compared to an oblivious tree that includes all invariants. Furthermore, it lacks a compelling practical use case demonstrating more efficient matching relative to other tree-based architectures, such as model merging, which can be used to improve performance. To strengthen the argument, it would be necessary to include some experiment similar to the one shown in Figure 8, applied to the proposed architecture.

**Questions:**

- In Figure 6, I don't quite understand why AM considering only permutation increases the barrier more than naive interpolation. Why does considering only activation result in a worse match than identity? The authors explain this in the main text as follows, but I would like a more detailed explanation.

    > This is because parameter values are not used during the rearrangement of the tree in AM.
    >
- Why do the authors think that the barrier after matching increases as depth increases? Ainsworth et al. (2023) showed that the barrier decreases as width increases, but they did not experiment with depth, and Entezari et al. (2022) measured the barrier for changes in width and depth without matching, which is different from the experiment in Figure 6, which measured the barrier after matching.
- Is there a compelling reason to use activation matching over weight matching, given that it does not seem to offer any clear advantages? Or is it simply intended to replicate the methodology of Ainsworth et al. (2023) for validation purposes?
- It would be useful to know the actual runtime required for matching. Compared to neural networks commonly considered in standard LMC research, such as ResNet, the number of parameters here is significantly lower. Given this, even with a large number of invariance patterns, it may not take very long to check all of them.
- Some minor suggestions:
    - In Figure 6, the x-axis represents different categories without a clear order, making the use of a line plot appear unnatural. A bar plot or box plot would be more appropriate in this case.
    - In the tables and figures, the notation "Ours" does not clearly give the intended meaning. While I understand that it stands for "Perm & Flip & Order", it is somewhat confusing. This is especially noticeable in Tables 3 and 4, where "Decision List (Modified)" is also the proposed method ("Ours"), which makes the absence of results in the corresponding cell seem unnatural. Furthermore, I assume that "Naive" is meant to indicate the results without matching, but this is not explained anywhere. To improve clarity, it would be helpful either to label it more explicitly as "naive interpolation", as in the cited papers, or to provide additional clarification in the main text.

---

> ### Author Response · Authors · 2024-11-21
>
> Thank you for your review.
>
> > The authors do not provide a more efficient matching algorithm other than exhaustive search.
>
> This is because the importance of considering additional invariants with an efficient (potentially heuristic) matching algorithm is relatively low, as discussed in the final paragraph of Section 4.2. There, we have noted that deep perfect binary trees are not particularly significant, which is supported both empirically and theoretically by the NTK theory [1], and such an efficient matching algorithm would only be necessary for handling those deep perfect binary trees. When exact solutions are not required and heuristic methods are acceptable, one possible approach could involve using a greedy algorithm. For example, to consider subtree flip invariance, this algorithm would sequentially determine the orientation of inequalities, starting from the root of the tree.
>
> > it would be necessary to include some experiment similar to the one shown in Figure 8, applied to the proposed architecture.
>
> Thank you for your suggestion. We have added experimental results on decision lists in the Appendix. Similar to the case of the perfect binary tree, we observed performance improvements when using model merging with the split dataset. Additionally, decision lists offer more efficient matching, which could be advantageous in certain practical situations.
>
> > In Figure 6, I don't quite understand why AM considering only permutation increases the
> barrier more than naive interpolation.
>
> When AM ignores subtree flip invariance, linear interpolation between parameter vectors oriented in opposite directions is likely to occur. As discussed in Section 3.1, subtree flip invariance involves reversing the signs of $w_{m,n}$ and $b_{m,n}$​ to flip inequality signs. Ignoring this can lead to interpolations between oppositely oriented parameters, driving the combined model parameters toward zero. We believe this is one of the reasons why the barrier becomes larger than that of naive interpolation.
>
> > Why do the authors think that the barrier after matching increases as depth increases?
>
> We think that the shape of the loss landscape, influenced by the model's structure, plays a role. As the depth of the tree increases, the loss landscape is likely to become more complex. If the loss valleys have irregular shapes, convergence within the same valley could still face larger barriers due to these distortions in the loss landscape. By contrast, in soft tree ensembles, increasing the number of trees tends to simplify the loss landscape, often making training easier. This phenomenon is partially explained by theories such as NTK [1]. These insights may help explain the observed results.
>
> > Is there a compelling reason to use activation matching over weight matching, given that it does not seem to offer any clear advantages? Or is it simply intended to replicate the methodology of Ainsworth et al. (2023) for validation purposes?
>
> Yes, we have investigated both AM and WM, not only to follow previous research but also because the theoretical validity of both AM and WM has been proven in [2].
>
> > It would be useful to know the actual runtime required for matching. Compared to neural networks commonly considered in standard LMC research, such as ResNet, the number of parameters here is significantly lower. Given this, even with a large number of invariance patterns, it may not take very long to check all of them.
>
> Thank you for your comment. While we agree that there is a potential for high-speed operation, our Python code for handling tree-structured data is not well optimized. As a result, the computational overhead required to prepare data for LAP can be large, making direct runtime comparisons challenging at this stage. Therefore, we focused on discussing computational complexity instead in our paper. With implementation optimizations, we believe it is possible to achieve comparable or even faster processing speeds depending on the number of invariance patterns involved. As noted in our paper, while multi-layer neural networks typically require performing LAP multiple times due to coordinate descent, this iterative operation is unnecessary in tree ensembles. This generally reduces the computational cost for tree ensembles.
>
> > Some minor suggestions
>
> Thank you for your suggestion. We have fixed Figure 6 and clarified the specific meanings of terms to make them explicit. To avoid any confusion, we have explicitly defined all the notations, "Naive," "Tree Permutation”, and "Ours", at the end of the first paragraph in Section 4.2.
>
> ----
>
> [1] Kanoh & Sugiyama, A Neural Tangent Kernel Perspective of Infinite Tree Ensembles, ICLR2022
>
> [2] Zhou et al, Going Beyond Linear Mode Connectivity: The Layerwise Linear Feature Connectivity, NeurIPS2023

---

> > ### Comment · Reviewer_y2Hs · 2024-11-25
> >
> > Thank you to the authors for the detailed responses. The authors have addressed my comments appropriately.

---

### Meta-Review · Area_Chair_tSq1 · 2024-12-20

**Metareview:**

The paper extends the results on linear mode connectivity (LMC) to differentiable tree ensembles. LMC was previously demonstrated only for neural network models  and this paper studies this phenomenon for a new class of models. In neural networks, accounting for permutation invariance in neurons is sufficient for LMC typically. The authors show that the analogous notion of permutation invariance does not lead to LMC in differentiable tree ensembles. However, accounting for a broader class of invariances (subtree flip, splitting order) is sufficient for LMC. The authors propose two matching methods: activation and weight matching; they then study LMC when accounting for different classes of invariance in different model classes and on different problems. The authors also provide results on model merging via LMC and propose decision lists, a variation of decision trees with reduced set of invariances.

Strenghts:
- The paper for the first time extends mode connectivity research beyond neural networks. Model merging is a potentially exciting application. Moreover, combinations of decision trees and neural networks (such as MoE transformer models) are highly practically relevant.
- The paper is well-written and the main results are easy to follow.
- The authors perform an extensive evaluation on a range of tabular tasks. During the rebuttal, the authors also added experiments on the MNIST dataset.
- The proposed methods work well empirically and achieve their goal (establishing LMC).

Weaknesses:
- No theoretical understanding of LMC in differentiable tree ensembles.
- Not completely clear how practically relevant these models are at the moment, and to what extent LMC is practically relevant. Model merging is potentially exciting, but it’s not clear if there are practical settings where it would be relevant.
- Similarly, not clear if the proposed decision lists architecture is practically relevant.

Decision recommendation: This is an original, interesting paper extending linear mode connectivity to a new class of models. While the practical relevance is not completely clear, I believe the paper makes a valuable contribution to LMC literature and recommend accepting it.

**Additional Comments On Reviewer Discussion:**

The reviewers are unanimous in accepting  the paper: 8, 8, 6. Reviewers raised detailed technical questions and concerns (no theory, no results on sparcification, utility of decision lists, results on image data, …), which were addressed during the rebuttal phase. Two of the reviewers acknowledged the rebuttal and decided to keep their scores.

---

### Decision · Program_Chairs · 2025-01-22

Accept (Spotlight)